



# α-Pinene secondary organic aerosol yields increase at higher relative humidity and low NOₓ conditions

Lisa Stirnweis[1,$], Claudia Marcolli[2,3], Josef Dommen[1], Peter Barmet[1,+], Carla Frege[1], Stephen M. Platt[1,§], Emily A. Bruns[1], Manuel Krapf[1], Jay G. Slowik[1], Robert Wolf[1], Andre S. H. Prévôt[1], Imad El-Haddad[1*], and Urs Baltensperger[1*]

[1]Laboratory of Atmospheric Chemistry, Paul Scherrer Institute, 5232 Villigen, Switzerland

[2]Institute for Atmospheric and Climate Science, ETH Zurich, 8092 Zurich, Switzerland

[3]Marcolli Chemistry and Physics Consulting GmbH, Zurich, Switzerland

[$]Now at: Federal Office for Radiation Protection, Dept. Radiation Protection and Environment, 85764 Oberschleissheim, Germany

[+]Now at: Department Construction, Traffic and Environment, Canton of Aargau, 5001 Aarau, Switzerland

[§]Now at: Department of Atmosphere and Climate, Norwegian Institute for Air Research, 2007 Kjeller, Norway

[*]*Correspondence to*: Imad El-Haddad (imad.el-haddad@psi.ch); Urs Baltensperger (urs.baltensperger@psi.ch)

**Abstract.** Secondary organic aerosol (SOA) yields from the photooxidation of α-pinene were investigated in smog chamber (SC) experiments at low (23-29 %) and high (60-69 %) relative humidity (RH), various NOₓ/VOC ratios (0.04-3.8) and with different aerosol seed chemical compositions (acidic to neutralized sulfate-containing or hydrophobic organic). A combination of a scanning mobility particle sizer and an Aerodyne high resolution time-of-flight aerosol mass spectrometer was used to determine SOA mass concentration and chemical composition. We present wall-loss-corrected yields as a function of absorptive masses combining organics and the bound liquid water content. High RH increased SOA yields by up to six times (1.5-6.4) compared to low RH. The yields at low NOₓ/VOC ratios were in general higher compared to yields at high NOₓ/VOC ratios. This NOₓ dependence follows the same trend as seen in previous studies for α-pinene SOA.

A novel approach of data evaluation using volatility distributions derived from experimental data served as basis for thermodynamic phase partitioning calculations of model mixtures in this study. These calculations predict liquid-liquid phase separation into organic-rich and electrolyte phases. At low NOₓ conditions, equilibrium partitioning between the gas and liquid phases can explain most of the increase in SOA yields at high RH. This is indicated by the model results, when in addition to the α-pinene photooxidation products described in the literature, more fragmented and oxidized organic compounds are added to the model mixtures. This increase is driven by both the increase in the absorptive mass due to the additional particulate water and the solution non-ideality described by the activity coefficients. In contrast, at high NOₓ, equilibrium partitioning alone could not explain the strong increase in the yields with increased RH. This suggests that other processes including the reactive uptake of semi-volatile species into the liquid phase may occur and be enhanced at higher RH, especially for compounds formed under high NOₓ conditions such as carbonyls.



## 1 Introduction

Organic aerosol (OA) accounts for 20 to 90 % of the submicron ambient aerosol (Jimenez et al., 2009 and references therein), a great part of which is secondary organic aerosol (SOA) formed via the condensation of oxidation products of gas-phase precursors. Several direct (e.g. radiocarbon dating) and indirect observations underline the key role of biogenic volatile organic compounds (VOC) for SOA formation (El Haddad et al., 2013 and references therein). Current state-of-the-art models are unable to predict the burden of biogenic SOA, especially in urban atmospheres (Hoyle et al., 2011), highlighting a fundamental deficit in our knowledge of the chemical pathways by which SOA accumulates and evolves in the atmosphere.

The ensemble of gaseous and particulate phase species involved in SOA formation is immensely complex. The compounds relevant for SOA formation are often a minor fraction, resulting in yields (SOA formed to precursor reacted) of only a few percent. Their chemical composition and volatility distribution strongly depend on the oxidation conditions, most notably on the fate of organic peroxy radicals ($RO_2$), which either react with nitrogen oxides ($NO_x$) or other peroxy radicals ($RO_2$ and $HO_2$). The influence of $NO_x$ on the oxidation mechanisms and SOA formation is commonly described by the $NO_x$/VOC ratio and has been under close scrutiny lately. For most light precursors, such as isoprene and monoterpenes (including α-pinene), SOA yields appear to be strongly influenced by the $NO_x$/VOC ratio, with a general enhancement observed under low $NO_x$ conditions (Ng et al., 2007; Presto et al., 2005).

Semi-volatile species involved in SOA formation are subject to ongoing chemical degradation, which may lead to compounds of either lower (when functionalization dominates) or higher volatility (when fragmentation dominates). As a consequence, SOA yields and degrees of oxygenation (described by the atomic oxygen to carbon ratio O:C) may depend on the extent to which these species were exposed to oxidants (Kroll and Seinfeld, 2008; Donahue et al., 2012a).

SOA yields are generally described by the absorptive equilibrium partitioning of condensable species to a well-mixed liquid phase (Odum et al., 1996), which depends upon the chemical species' saturation vapor pressures (classically two model products are considered) and their liquid-phase activities (modified Raoult's law). Donahue and co-workers proposed the use of a "volatility basis set" (VBS) for a better representation of the wide range of OA in the atmosphere and the ongoing oxidation of semi-volatile organics (Donahue et al., 2011; 2012b and references therein). However, several difficulties remain in estimating or measuring the saturation vapor pressures (e.g. Huisman et al., 2013; Bilde et al., 2015), activity coefficients and the mean molecular weight of the condensing species (e.g. Clegg et al., 2008a, b).

Difficulties increase when considering the role of relative humidity (RH) and of electrolyte particles on organic partitioning (Zuend and Seinfeld, 2013 and references therein). From a thermodynamic point of view, water interacts with SOA components by altering the water content of aerosol particles (also at subsaturated conditions) and hence the equilibrium concentration of water-soluble organic compounds. According to the equilibrium equation of Pankow (1994), an increase in the condensed fraction of the organics can be achieved by 1) increasing the absorptive particulate mass, 2) decreasing the average molecular weight of condensed species or 3) decreasing the activity coefficients of organic species (Pankow, 1994). Therefore, it is expected that an increase in particulate water content would in principle enhance SOA yields of water-miscible species. Model calculations predict a pronounced effect, especially at low organic mass loadings (Pankow, 2010). However,





literature reports based on experimental results seem contradictory: RH dependent yields have been reported for the ozonolysis of limonene, α-pinene and $\Delta^3$-carene (Jonsson et al., 2006), while Prisle et al. (2010) found a negligible impact of RH on SOA yields from α-pinene ozonolysis. A substantial effect of RH on SOA yields was observed only for studies at precursor concentrations <1000 ppbv and with large variation in RH (0.01 % &

31 % RH (Bonn et al., 2002); < 2–58 % RH (Cocker et al., 2001); < 2–85 % RH (Jonsson et al., 2006).

The impact of RH on the partitioning of organic species may be severely suppressed by considerable deviations from ideal mixing between the condensing organic species and the prevailing condensed phase. A growing number of studies show that organic compounds are salted out in internally mixed organic/ inorganic/ water aerosol particles and two stable liquid phases may develop: an aqueous electrolyte solution and an organic

solution (Marcolli and Krieger, 2006; You et al., 2014). The miscibility of an organic compound in aqueous droplets containing electrolytes depends on numerous factors including temperature, relative humidity, organic compound polarity, relative contribution of the compound to the bulk particulate matter and the chemical nature of the electrolytes (You et al., 2013; Zuend and Seinfeld, 2013). For example, phase separation was observed to always occur for organic compounds with an O:C < 0.5 and at low relative humidity (You et al., 2013), and is

especially pronounced in the case of ammonium sulfate (compared to ammonium hydrogen sulfate and nitrate). While neglecting phase separation of organic compounds and the effect of RH thereon bears the potential for invalid yield predictions (e.g. if the condensed phase is considered to comprise a mixed electrolyte and organic solution), the way the complex organic matrix interacts with water and the inorganic species remains virtually unknown.

Another complication that might influence the interpretation of chamber experiments conducted at different RH is the enhancement of SOA yields by the potential reactive uptake of organic products into the particle phase (Kroll and Seinfeld, 2008; Kleindienst et al., 2006; Jang et al., 2002; Iinuma et al., 2007). The mechanisms by which such reactions occur are not fully identified, but may involve an ester or an aldol formation, which are expected to be favourable in the absence of water and are possibly catalytic under acidic conditions. Assessing

the relative importance of particle phase processing at different particle water contents would require decoupling SOA thermodynamics and additional reactivity.

In this study, we examine the impact of particle water content on the chemical composition and yields of α-pinene SOA formed under low and high NOx. This is performed by varying NOx/α-pinene ratios, aerosol seed composition (hygroscopicity, acidity) and relative humidity. Results are parameterized within a thermodynamic

framework to investigate whether changes in SOA non-ideal mixing properties with particle water content may explain the variation in SOA yields with RH. SOA yields reported here may aid the parameterization of the NOx and particulate water dependence of α-pinene SOA production for further use in atmospheric models.





## 2    Methods

### 2.1    Experimental setup and instrumentation

20 experiments, listed in Table 1, were carried out in the smog chamber (SC) of the Paul Scherrer Institute (PSI):
a Teflon bag of 27 m³ suspended in a temperature-controlled housing (Paulsen et al., 2005). Photochemistry was
initiated by four xenon arc lamps (4 kW rated power, $1.55 \cdot 10^5$ lumens each, XBO 4000 W/HS, OSRAM), facing
parallel to the SC bag, and emitting a light spectrum similar to the solar spectrum, and 80 black lights (Philips,
Cleo performance 100 W) to accelerate the aging process, located underneath the SC bag, with emission between
300–400 nm wavelength (light characterisation in Platt et al. (2013)). A reflecting aluminum foil surrounds the
SC bag to maintain light intensity and light diffusion.

Various parameters were monitored in the SC. The temperature ($T$) and RH measurement was optimized by
passing SC air through a radiation shielded sensor. One of two different high resolution time-of-flight aerosol
mass spectrometers (HR-ToF-AMS, Aerodyne Research, Inc., Billerica, MA, USA) was operated during three
different campaigns to measure online size-resolved chemical composition (organics, ammonium, nitrate,
sulfate, chloride) of non-refractory particles (DeCarlo et al., 2006). The HR-ToF-AMS were equipped with two
different $PM_{2.5}$ lenses (Williams et al., 2013) to sample particles up to large diameters above 1 µm. The sampled
aerosol was dried (~10 % RH) before measurement. A supporting flow of ~1.5 L min$^{-1}$ was maintained parallel
to the HR-ToF-AMS to minimize diffusive losses in the sampling lines. The HR-ToF-AMS data was corrected
for gas-phase contributions.

Two scanning mobility particle sizers (SMPS) were additionally deployed for the measurement of the aerosol
size distributions. The first SMPS (a custom built differential mobility analyzer, DMA: extended length $L_{eff} =$
0.93 cm, $d_{m\,max}$=1000 nm, recirculating sheath flow, and a condensation particle counter, CPC 3022 (TSI)) was
connected to the HR-ToF-AMS sampling line to analyze the dried particles. A second SMPS (SMPS$_{wet}$, a TSI
DMA classifier 3081 with recirculating sheath flow and a TSI CPC 3022A) and a CPC (TSI: CPC 3025A)
measured the wet particle number size distribution and total number concentration ($d > 3$ nm), respectively.

Gas-phase compounds with a higher proton affinity than water (166.5 kcal mol$^{-1}$) were measured with a
quadrupole proton transfer reaction mass spectrometer (PTR-MS, Ionicon). The PTR-MS was calibrated before
each experiment for α-pinene, detected at $m/z$ 137 and $m/z$ 81; the accuracy of these measurements was estimated
to be ~5%, based on the purity indicated on the calibration gas cylinder.

A modified NO$_x$ instrument including a photolytic NO$_2$-to-NO converter (Thermo Environmental Instruments
42C trace level NO$_x$ analyzer equipped with a blue light converter) and two ozone monitors (Monitor Labs 8810
ozone analyzer, Environics S300 ozone analyzer) monitored NO$_x$ and O$_3$ in the chamber.

### 2.2    Chamber operation and aerosol seeding

SOA formation and growth from α-pinene was induced by the following SC operation sequence: (1)
humidification of the chamber, (2) addition of seed aerosol, (3) introduction of VOCs, (4) addition of nitrous
acid (HONO) as an OH precursor, (5) addition of nitrogen oxides (equal amounts of NO + NO$_2$) if applicable,
(6) an equilibration period (30–45 min), (7) switching on of xenon and black lights to generate OH radicals, and
(8) a reaction time of 5 to 20 h (corresponding to $0-2 \times 10^8$ cm$^{-3}$ h OH exposure, see Sect. 2.4). Experimental
conditions for each individual experiment are summarized in Table 1. Ahead of each experiment, cleaning of the





SC was performed by the injection of several ppmv of ozone (5 h) and the simultaneous irradiation with black lights (10 h) at a temperature of 20 °C. This was followed by a pure air flushing period at high relative humidity (~60 %) at a temperature of approximately 30 °C for at least 20 h. Three blank experiments (seed aerosol, lights switched on, high RH, but without adding α-pinene) were carried out to make sure that the organic aerosol
formed during the experiments is not significantly influenced by background contamination in the SC. The organic mass concentration formed was substantially lower (< 0.1 up to 2.8 µg m$^{-3}$) than during comparable experiments (similar $NO_x$/VOC and RH).

In the chamber, the temperature varied between 21 °C and 26 °C. Due to heat from the xenon and black lights, the temperature increased, stabilizing only ~1 h after experiment start. The increase in temperature of 1–4 °C led
to an absolute decrease in RH of ~2-20 %, thus the RH range in Table 1, 23–67 %, is given for the temperature-stable period. α-Pinene (98 %, Aldrich) and an OH reactivity tracer (9-times deuterated butanol, 98 %, D9, Cambridge Isotope Laboratories), hereafter referred to as butanol-d9 (1 µL injected ≈ 10 ppbv in the SC), were sequentially injected into an evaporation glass bulb heated to 80 °C. The two VOCs were transferred into the bag by a dilution and flush flow (each 15 L min$^{-1}$, maintained for 15 min) from an air purifier (737-250 series,
AADCO Instruments, Inc., USA), further referred to as "pure air". Initial α-pinene concentrations were 16.1–31.7 ppbv.

HONO was used as a source of both NO and OH, produced by continuous mixing in a reaction vessel of the reagents sodium nitrite ($NaNO_2$, 1 mmol L$^{-1}$ in milliQ-$H_2O$) and three different concentrations of sulfuric acid solutions ($H_2SO_4$, 1 mmol L$^{-1}$ (expts. 1-6), 10 mmol L$^{-1}$ (expts. 7, 8, 11, 13) and 100 mmol L$^{-1}$ (expts. 9, 10, 12,
14) in milliQ-$H_2O$) (Taira and Kanda, 1990). HONO was carried by 2.5 ± 0.2 L min$^{-1}$ pure air flow into the SC. 2 ppbv (± 10 %) of HONO were injected before lights on to initiate photochemistry and the injection was continued throughout all experiments. A chemiluminescence-based $NO_x$ instrument (Monitor Labs 9841A $NO_x$ analyzer) was attached to the HONO source to monitor the injected concentration throughout the experiment. In addition, equal concentrations of NO (99.8 %; 1005 ppmv ± 2 %) and $NO_2$ (purity: 98 %; 1005 ppmv ± 3 %),
resulting in 19.6–75.1 ppbv initial $NO_x$, were added during experiments with $NO_x$/α-pinene > 1. Within the results section, the two terms "low $NO_x$" and "high $NO_x$" refer to the following conditions:

- low $NO_x$ = $NO_x$/α-pinene < 0.1, with continuous HONO injection, indicated by an asterisk in Table 1 and figures.
- high $NO_x$ = $NO_x$/α-pinene > 1: Initial injection of NO + $NO_2$ with continuous HONO injection.

In Table 1, we report for high $NO_x$ conditions the initial $NO_x$ concentration, (which decays with time) and for low $NO_x$ conditions the mean $NO_x$ concentration.

During 14 experiments (no. 1–14) and three blank experiments, an ammonium hydrogen sulfate ($NH_4HSO_4$, Aldrich) solution in ultrapure Milli-Q water (1 g L$^{-1}$) was nebulized (0.6 L min$^{-1}$) and introduced into the SC with a pure air dilution flow of 10 L min$^{-1}$ to act as seed particles. To keep the seed aerosol in a liquid state, no
drier was used behind the nebulizer. We determined the acidity of the seed particles, here described by the ratio $NH_4$/$SO_4$, by the comparison between the HR-ToF-AMS measurements of the seed particles and nebulized ($NH_4$)$_2SO_4$ and $NH_4HSO_4$ solutions. An $NH_4$/$SO_4$ ratio between 1 and 2 indicates a rather neutral seed composition, representing a mixture of $NH_4HSO_4$, ($NH_4$)$_3$($SO_4$)$_2$ (letovicite) and ($NH_4$)$_2SO_4$. For simplicity, we



replaced $(NH_4)_3(SO_4)_2$ by an equal mix of $NH_4HSO_4$ and $(NH_4)_2SO_4$ for the assumption of density and growth factors. By contrast, an $NH_4/SO_4 \leq 1$ indicates an acidic seed, consisting of $NH_4HSO_4$ and $H_2SO_4$.

During experiments 7–14, the nebulized $NH_4HSO_4$ solution was partly neutralized to $(NH_4)_2SO_4$, presumably by background $NH_3$ (Fig. S1, Supplement). During experiments 1–6 the seed was composed of an aqueous mixture of $NH_4HSO_4$ and $H_2SO_4$, due to the in–situ production of gas-phase $H_2SO_4$ via the HONO injection system, suppressing the seed neutralization by $NH_3$ (even though particulate $H_2SO_4$ was minimised by a teflon filter applied between the HONO source and the SC).

Additionally, three α-pinene experiments (no. 15–17) and one blank experiment were conducted using an inert hydrophobic fluorinated hydrocarbon seed (further referred to as CF-seed; $CF_3CF_2CF_2O$-[CF$(CF_3)CF_2$-O-]$_n CF_2CF_3$; Krytox® 1525). The CF-seed was generated via the evaporation of the pure compound at a temperature of 125–145 °C and subsequent homogeneous nucleation in a pure air flow of $2.4 \pm 0.1$ L min$^{-1}$. The CF-seed concentration was 6.7–10 µg m$^{-3}$ when lights were switched on, and decayed very rapidly to values below detection limit of the HR-ToF-AMS (0.1 µg m$^{-3}$) during the course of the experiment. The CF-seed mass spectrum in the HR-ToF-AMS is clearly distinct from that of α-pinene SOA, with main contributions at *m/z* 69 (CF$_3$), *m/z* 169 (C$_3$F$_7$) and *m/z* 119 (C$_2$F$_5$) (Fig. S2 and Table S1, Supplement).

Table 1 lists the expected physical state and seed composition of each experiment dependent on RH (AHS = ammonium hydrogen sulfate ($NH_4HSO_4$); AS = ammonium sulfate ($(NH_4)_2SO_4$); SA = sulfuric acid ($H_2SO_4$); CF = fluorinated carbon. Submicrometer AS particles and letovicite particles ($(NH_4)_3H(SO_4)_2$) are expected to effloresce at about 35 % RH while more acidic particles should remain liquid between 20 and 30 % RH (Martin, 2000; Ciobanu et al., 2010).

### 2.3 Estimation of the hygroscopic growth factors and liquid water content

The absolute liquid water content (LWC) of the aerosol particles was derived for the bulk aerosol mass and the size resolved mass distribution, based on literature growth factors, the measured RH and chemical composition. The growth factor GF (RH) of a particle is defined as the ratio of the wet diameter at a given RH to the dry diameter: $GF(RH) = d(RH)/d_{dry}$. Inorganic GFs were taken from the Aerosol Diameter Dependent Equilibrium Model (ADDEM, Topping et al., 2005) for diameters of 360 nm. Organic GFs were derived using the relationship between the hygroscopicity parameter $\kappa$ and the degree of oxygenation [$\kappa = 0.29 \times$(O:C)] from Chang et al. (2010), well representing the hygroscopicity of α-pinene SOA measured by Massoli et al. (2010). The measured degree of oxygenation at an OH exposure of $(2.0 \pm 0.5) \times 10^7$ cm$^{-3}$ h was used to derive $\kappa$, which in turn was converted to GF, assuming a negligible curvature (Kelvin) effect (Kreidenweis et al., 2005):

$$GF(RH) = \left(1 + \kappa \frac{\frac{RH}{100\%}}{1 - \frac{RH}{100\%}}\right)^{\frac{1}{3}} \qquad (1)$$

The mixed GFs for aerosol containing inorganic and organic species were determined using as a first approximation the Zdanovskii-Stokes-Robinson (ZSR) volume mixing rule (Stokes and Robinson, 1966):

$$GF_{mixed}(RH) = \left(\sum_i \varepsilon_i \times \left(GF_i(RH)\right)^3\right)^{\frac{1}{3}} \qquad (2)$$

where $\varepsilon_i$ and GF$_i$(RH) denote the volume fraction and GF (RH) of species *i*, respectively. The H$_2$O volume ($V_{H_2O}$) was calculated using the definition of GF (RH) and the dry volume ($V_{dry}$):





$$V_{H_2O} = V_{dry} \times (GF(RH)^3 - 1) \qquad (3)$$

$V_{H_2O}$ multiplied by the density of water (1 g cm$^{-3}$) results in the LWC.

The dry ($S_{dry}$) and wet ($S_{wet}$) surfaces from HR-ToF-AMS size resolved data were calculated with Eq. (4) and Eq. (5), respectively:

$$S_{dry} = 6 \times (V_{dry})/d_{dry} \qquad (4)$$

$$S_{wet} = 6 \times (V_{dry} + V_{H_2O})/(d(RH)) \qquad (5)$$

The LWC and surface distributions were calculated using size resolved pToF (particle time-of-flight) data of the HR-ToF-AMS. Due to the low pToF signal of NH$_4$, the NH$_4$ surface distributions were estimated based on SO$_4$ pToF measurements.

**2.4      Determination of OH exposure and extent of α-pinene ozonolysis**

The gas-phase composition, the OH concentration and the photochemical age of a chemical reaction system may considerably differ between experiments of the same duration. Furthermore, variation of the OH concentration within a single experiment means that the photochemical age is not necessarily directly proportional to the light exposure time. Consequently, we discuss reaction time in terms of OH exposure (molecules cm$^{-3}$ h), defined as

the OH concentration integrated over time. OH exposures were derived based on the decay of the OH tracer butanol-d9, detected by the PTR-MS as M+H$^+$-H$_2$O at $m/z$ 66), following the methodology introduced by Barmet et al. (2012). The OH exposure was determined by the integration of the following expression:

$$OH\,exposure = -\int_{t_1=0}^{t} \left( \frac{1}{k_{OH,butanol-d9}} \cdot \left( \frac{\Delta \ln(butanol-d9)}{\Delta t} + \frac{f_{dil}}{V} \right) \right) dt \qquad (6)$$

where butanol-d9 and $k_{OH,\,butanol-d9}$ (= $3.4 \times 10^{-12}$ cm$^3$ molecules$^{-1}$ s$^{-1}$) are the butanol-d9 concentration and its

reaction rate constant against OH, respectively, $t$ is the time after lights on, $V$ the chamber volume (we assume as a first approximation a constant chamber volume of 27 m$^3$), and $f_{dil}$ the dilution flow due to HONO input (Sect. 2.2).

The percentage of α-pinene reacted with OH and O$_3$ was derived based on Eq. (7):

$$-\frac{d(\alpha-pin)}{dt} = k_{O_3,\alpha-pin}\,[O_3] \times [\alpha - pin] + k_{OH,\alpha-pin}\,[OH] \times [\alpha - pin] + \frac{f_{dil}}{V} \qquad (7)$$

where [α-pin], [O$_3$] and [OH] denote the concentrations of α-pinene (measured by the PTR-MS at $m/z$ 137 and 81), O$_3$ and OH, respectively, and $k_{OH,\,\alpha-pin}$ (= $5.3 \times 10^{-11}$ cm$^3$ molecules$^{-1}$ s$^{-1}$) and $k_{O3,\,\alpha-pin}$ (= $8.9 \times 10^{-17}$ cm$^3$ molecules$^{-1}$ s$^{-1}$) the reaction rate constants of α-pinene with OH and O$_3$, respectively. The production of O$_3$ was faster under high NO$_x$ compared to low NO$_x$, due to an efficient VOC-NO$_x$ catalytic cycle. Therefore, the lowest percentage (65 %) of α-pinene reacted with OH was achieved during experiment no. 9, under high NO$_x$

conditions (for all experiments, the percentage of α-pinene reacted with OH ranged between 65–88%, Table 1). We also expect that the further processing of the first generation products formed via α-pinene reaction with OH or O$_3$ – which don't contain C=C bonds – to predominantly proceed through reactions with OH. Accordingly, we conclude that SOA compounds detected under our conditions are mainly from OH chemistry.





### 2.5 Determination of suspended and wall-loss-corrected organic masses and yields

***Suspended OA mass.*** The suspended organic mass concentration $C_{OA}^{sus}$ was derived by utilizing the chemical composition measurements from the HR-ToF-AMS scaled to the total volume measured by the SMPS (Fig. S3 and Fig. S4, Supplement), using compound-specific densities ($\rho_{org}$ = 1.4 g cm⁻³, $\rho_{NH4HSO4}$ = 1.79 g cm⁻³, $\rho_{(NH4)2SO4}$

= 1.77 g cm⁻³, $\rho_{H2SO4}$ = 1.83 g cm⁻³).

For some experiments (9, 10, 12 and 14-17), the organic mass concentrations determined by the HR-ToF-AMS were corrected for a sub-unity transmission efficiency at the lower edge cut-off of one of the two PM$_{2.5}$ lenses employed (Fig. S3 and Fig. S4, Supplement). Additionally, for organic mass calculation, we assumed that the measured NO$_3$ signals are entirely related to organonitrates (RONO$_2$), rather than NH$_4$NO$_3$. This assumption

mainly stems from (1) the observation of the NO$_3$ signal in the same particle size region as OA rather than SO$_4^{2-}$ (see size-resolved pToF data, Sect. 3.3), while inorganic nitrate would be expected to mix within an electrolyte rich aerosol and (2) the presence of NO$_3$ under acidic conditions, which are thermodynamically unfavorable for the partitioning of nitric acid. This is also supported by the higher NO$^+$/NO$_2^+$ ratios measured in the SC compared to ratios recorded during NH$_4$NO$_3$ nebulization (on average 1–2.8 times higher, Supplement Fig. S5),

typically expected from organonitrates (Farmer et al., 2010). We cannot exclude that a part of the NO$_3$ signal originates from NH$_4$NO$_3$. However, even attributing all detected nitrate to NH$_4$NO$_3$ would increase the calculated LWC by 1–13 % and decrease the calculated OA mass by 2–7 % only, which would not influence our conclusions. Finally, for yield calculations we assume the accuracy of the aerosol phase measurements to be 30% (Canagaratna et al., 2007).

***Wall-loss-corrected OA mass.*** To obtain the total $C_{OA}$ concentration corrected for losses of particles and vapors to the chamber walls, we use Eq. (8), introduced by Hildebrandt et al. (2011), based on the mass balances of the suspended organic aerosol mass, $C_{OA}^{sus}$, and the mass of the organic aerosols on the walls, $C_{OA}^{walls}$ (summed up to derive $C_{OA}$):

$$\frac{d}{dt}\left[C_{OA}^{walls}(t)\right] = k_{OA}^{w}(t)C_{OA}^{sus}(t) + \omega(t)\cdot\left(k_{OA}^{w}(t)C_{OA}^{sus}(t) + \frac{d}{dt}\left[C_{OA}^{sus}(t)\right]\right)\cdot\frac{C_{OA}^{walls}(t)}{C_{OA}^{sus}(t)} \qquad (8)$$

Here, $k_{OA}^{w}$ represents the loss rate constant of organic particles to the walls, derived by fitting the suspended organic mass concentration 5–8 h after lights were switched on in the SC, when SOA production is expected to be negligible (α-Pinene concentration < 1 ppbv, Fig. S6 and Table S2, Supplement). We determine an average loss rate of 0.13 µg m⁻³ h⁻¹, corresponding to a particle half-life of 5.3 h. The average $k_{OA}^{w}$ was used in the case of insufficient statistics to perform accurate fitting. In Eq. (8), ω, ranging between 0 and 1, is a dimensionless

proportionality coefficient between the mass of organic vapors that partition onto the wall-deposited particles and the mass of organic vapors that partition onto the suspended particles. Here, we neglect the condensation of organic vapors onto the wall-deposited particles, i.e. ω = 0, consistent with previous studies of α-pinene SOA production (Hildebrandt et al., 2011 and references therein). This assumption gives a lowest estimate of SOA yields, but does not influence the comparison between the experiments. Considering the second limiting case

ω = 1, i.e. an equal partitioning of organic vapors between the wall-deposited and suspended particles, would increase the determined SOA yields (by up to 40 %, and by 20 % on average).

Eq. (8) does not take the loss of SOA-forming vapors onto the clean Teflon walls of the chamber into consideration, which depends on the wall-to-seed surface ratio and may greatly suppress SOA yields from



laboratory chambers under certain conditions. We believe though that gas-wall partitioning does not significantly influence the interpretation of our results, focused on relative differences in the yields determined under different conditions. It is also worthwhile to note that (1) this effect was found to be minor for the α-pinene SOA system where SOA formation is dominated by quasi-equilibrium growth (Zhang et al., 2014; Nah et al., 2016), (2) SOA

mass formation rapidly evolved after lights were switched on and (3) we maintained a relatively constant wall-to-seed surface ratio for all experiments. Hence losses of organic vapors may lead to a systematic negative bias in the determined yields, but do not influence the comparison between the experiments.

*SOA yields determination and parameterization.* SOA mass yields, $Y$ (dimensionless quantity), are calculated from Eq. (9), as the organic mass concentration formed, $C_{OA}$, per precursor mass consumed, $\Delta C_{\alpha\text{-pinene}}$:

$$Y = \frac{C_{OA}}{\Delta C_{\alpha-pinene}} \tag{9}$$

For comparison purposes, $Y$ values are reported in Figure 2 and Table 1 and discussed in Sect. 3 at an OH exposure of $(2.0 \pm 0.5) \times 10^{7}$ cm$^{-3}$ h, reached during all experiments. The parameterization of smog chamber SOA yields measured is based on the absorptive partitioning theory of Pankow (Eq. (10)), which expresses the production of a set of semi-volatile surrogate products (total number $N$) as a function of the mass yield of these

products, $\alpha_i$, and their partitioning coefficients, $\xi_i$ (a dimensionless quantity reflecting the condensed-phase mass fraction of these products). The critical parameters driving the partitioning of these products are their effective saturation concentration, $C_i^*$, and the total concentration of the absorptive organic phase, $C_{OA}$. As discussed below, we consider the absorptive organic mass as the sum of the total OA concentration and the liquid water in this phase (Sect. 3.3).

$$Y = \sum_i^N \alpha_i \xi_i = \sum_i^N \alpha_i \left(1 + \frac{C_i^*}{C_{OA}}\right)^{-1} \tag{10}$$

In Eq. (10), $C_i^*$ (in µg m$^{-3}$) is a semi-empirical property (inverse of the Pankow-type partitioning coefficient, $K_{P,i}$) reflecting the saturation vapor pressure of the pure constituents $\left(p_{L,i}^o\right)$ and the way they interact with the organic mixture (effectively including liquid phase activity coefficients, $\gamma_i$), as expressed in Eq. (11):

$$C_i^* = \frac{10^6 M_i \gamma_i p_{L,i}^o}{760 RT} \tag{11}$$

Here, $M_i$ denotes the compound molecular weight, $R$ the ideal gas constant and $T$ the temperature. Smog chamber yields from single experiments are fitted as a function of $C_{OA}$ using the volatility basis set (VBS) (Donahue et al., 2006), which separates semi-volatile organics into logarithmically spaced bins of effective saturation concentrations $C_i^*$. Figure 5 shows the resulting parameterizations (lines) in comparison to the measured data (symbols) for each experiment.

To determine the yields per volatility bin ($\alpha_i$), wall-loss-corrected SOA yields, $Y$ as a function of wall-loss-corrected absorptive mass concentration $C_{OA}$ (data presented in Figure 5) were used. We assumed a total number of 5 bins: $N = 5$ with $C_i^* = 0.01, 0.1, 1, 10$ and $100$ µg m$^{-3}$. To solve Eq. (10) for the parameters $\alpha_i$, we introduced a novel approach using a Monte Carlo simulation. This approach provides best estimates of $\alpha_i$ values (data shown in Figure 6 and Fig. S7 in the Supplement), together with a measure for the uncertainties related to the

determination of the volatility distributions from SC experiments. The calculation proceeded as follows:





1/ From the calculated yields ($Y$), lower ($Y_1$) and upper ($Y_2$) yield curves were determined based on the estimated measurement accuracy of $C_{OA}$ and α-pinene mass. A possible yield domain was inscribed against $C_{OA}$ by plotting $Y_1$ and $Y_2$ *vs.* their corresponding lower and upper $C_{OA}$, respectively. The parameterized yield curves are shown in Figure 5.

2/ A range of possible inputs was defined for each of the parameters $\alpha_i$. This range is restricted within the following interval [0; 2*($Y_i$ - $Y_{i-1}$)]. This step was only necessary for computational reasons.

3/ $\alpha_i$-parameters were randomly generated over the defined intervals.

4/ Deterministic computations of $Y$ vs. $C_{OA}$ were performed using Eq. (10) and the generated $\alpha_i$ inputs.

5/ Volatility distributions that fell within the domain defined in step (1) were retained, aggregated and presented
as probability distribution functions for each of the five effective saturation concentrations $C_i^*$ (probability density function, PDF, Figure 6 and Fig. S7 in the Supplement).

### 2.6    Phase partitioning calculations

We introduce a novel approach where the volatility distributions derived from experimental data serve as basis for thermodynamic phase partitioning calculations of model mixtures. The methods developed by Zuend et al.
(2008; 2010; 2012) and Zuend and Seinfeld (2013) were used to calculate gas-particle and liquid-liquid phase partitioning. In the model, we assumed instantaneous reversible absorptive equilibrium of semi-volatile organic species into ideal and non-ideal liquid phase aerosols. To calculate activity coefficients of the organic species as a function of the liquid particle phase composition, the thermodynamic group-contribution model AIOMFAC (Aerosol Inorganic-Organic Mixtures Functional groups Activity Coefficients) developed by Zuend et al. (2008;
2011) was utilized. Positive and negative deviations of mole fraction-based activity coefficients from a value of unity (ideal mixing) indicate the degree of non-ideality in a mixture. The overall loading of the smog chamber was used to calculate the seed concentrations in µg m$^{-3}$, which was then transformed in moles of seed per volume (mol m$^{-3}$) assuming equal shares of AHS and SA for acidic seeds and equal shares of AHS and AS for neutral seeds. Because interaction parameters of some organic functional groups with $HSO_4^-$ are missing in AIOMFAC,
we assumed the interactions of organic compounds with $HSO_4^-$ and with $SO_4^{2-}$ to be similar. For all computations, metastable supersaturated salt solutions were allowed. Model calculations were performed at an OH exposure of $(2.0 \pm 0.5) \times 10^7$ cm$^{-3}$ h. Based on Eq. (10), SOA partitioning is driven by the compounds' volatility distributions, which depends on the compounds' effective saturation concentrations ($C_i^*$) and their relative abundance ($\alpha_i$).

*Simulated cases.* The following simulations were performed:

1/    Case org: non-ideal partitioning including liquid-liquid phase separation, LLPS (activity coefficients calculated with AIOMFAC) of the organic compounds between gas phase and a purely organic aerosol phase neglecting the presence of the seed aerosol and using reported model compounds only;

2/    Case id: ideal partitioning (activity coefficients all set to unity, LLPS cannot occur) between gas phase
and organic aerosol phase using reported model compounds only;

3/    Case sd: non-ideal partitioning including the seed aerosol to an internally mixed organic/AS aerosol using reported model compounds only;





4/ Case sdfr: non-ideal partitioning including the seed aerosol to an internally mixed organic/AS aerosol including the formation of fragmented oxidation products (see below);

5/ Case orgfr: non-ideal partitioning including LLPS to a purely organic aerosol phase neglecting the presence of the seed including the formation of fragmented oxidation products.

$\alpha_i$-parameters determined through the Monte-Carlo simulations assume that the absorptive mass consists of the organic phase (in accordance with assumptions in chemical transport models). This assumption is violated for cases sd and sdfr as compounds may partition into the inorganic phase. Nevertheless, the results obtained for these cases may still be examined in relative terms to inspect the effect of RH on SOA yields in the presence of an inorganic seed and the partitioning of SOA compounds between the inorganic and organic phases.

***Model compounds.*** To simulate SOA partitioning in AIOMFAC, $\alpha$-pinene photooxidation products reported in the literature (Eddingsaas et al., 2012; Jaoui and Kamens, 2001; Kleindienst et al., 2007; Valorso et al., 2011) were chosen as model compounds for cases org, id and sd, namely ValT4N10 (10[th] compound in Table 4 from Valorso et al. (2011)), 3-hydroxyglutaric acid, pinic acid, hopinonic acid, norpinic acid, 2-hydroxyterpenylic acid, 10-oxopinonic acid, and 4-oxopinonic acid. These compounds are listed in Table S4 in the Supplement

together with their relevant physicochemical properties (MW, O:C ratio, vapor pressures and chemical structures). For cases sdfr and orgfr additional oxidized fragmented products (3-oxoadipic acid, glutaric acid, 5-COOH-3-OH-pentanal, and succinic acid) were included (Table S4, Supplement). While these compounds were not reported to derive from $\alpha$-pinene oxidation, their structure, including carbon and oxygen numbers are very similar to the most abundant compounds detected by Chhabra et al. (2015) and Mutzel et al. (2015) using

chemical ionization mass spectrometry. Volatility distributions could reliably be determined for volatility bins $C* = 0.1\text{-}100$ µg m$^{-3}$. Nevertheless, also lower volatility products with $C* = 0.01$ µg m$^{-3}$ are formed. Therefore, this bin was loaded for all phase partitioning calculations with equal fractions of the model compounds diaterpenylic acid acetate, 3-MBTCA and ValT4N9 with mass yields of $10^{-5}$ each. This low mass fraction does not influence the organic yield but proved to aid the convergence of the phase partitioning calculation. To

achieve mass closure in the model, pinonaldehyde (MW: 168 g mol$^{-1}$) was assumed to represent the more volatile products, which do not partition to the condensed phase. Because pinonaldehyde resides almost totally in the gas phase, it was not explicitly modelled.

***Model compounds $C_j^*$ calculation.*** Model compounds $j$ were assigned to the volatility bins based on the calculated $C_j^*$ values using Eq. (11) and assuming ideal mixing ($\gamma_j = 1$). Using the actual activity coefficients

calculated with AIOMFAC is not possible because they are a result of the phase partitioning calculation. However, because activity coefficients proved to be generally in the range of 0.1 to 10, they did not alter the initial product assignments to the volatility bins which cover one order of magnitude in effective saturation concentration. Therefore, the initial allocation of the compounds to the volatility bins in the VBS remains valid after taking the non-ideality into account. $p_{L,j}^o$ used in the calculations were vapor pressures of pure compounds

in liquid or subcooled liquid state, at 298 K estimated using EVAPORATION (estimation of vapor pressure of organics, accounting for temperature, intramolecular, and non-additivity effects, (Compernolle et al., 2011)), without using the empirical correction term for functionalized diacids.

***Setting model compounds' relative abundances in the model.*** From the $\alpha_i$-parameters generated through the Monte Carlo simulations in Sect. 2.5, 10 sets per experiment were randomly selected for the phase partitioning




calculations (provided that these parameters fall within the 10[th] and the 90[th] percentiles, to avoid outliers). The 11 chosen experiments exclude experiments 15-17 with hydrophobic seeds, experiment 6 which has no counterpart experiment at high RH and experiments 5 and 14 where the COA concentration range was very limited to accurately derive $\alpha_i$-parameters (Sect. 2.5). The chosen parameters are listed in Table S5 in the Supplement. As several model compounds are assigned to a volatility bin $i$, the yield of a compound $j$ is expressed as its relative contribution within the volatility bin $i$, $\chi_{j,i}$, times the relative abundance of the bin $\alpha_i$. For each experiment and each simulated case the fitted $\chi_{j,i}$ values are listed in Tables S6-S8 in the Supplement. For case org the $\chi_{j,i}$-values were optimized to match the experimental organic yields and the measured O:C at the actual RH of the experiment. RH was then changed in the model to the value of the corresponding experiment (same experimental conditions but different RH) and the effects of RH on SOA yields and on degrees of oxygenation (O:C ratios) were then evaluated. For cases id and sd the same $\chi_{j,i}$, were used as for case org. For cases sdfr and orgfr, more fragmented compounds were added and their $\chi_{j,i}$-values were optimized to achieve agreement between measured and modelled organic mass yields and O:C ratios. Likewise, the effects of RH on SOA yields and on O:C ratios were then evaluated by changing the RH in the model.

The following pairs of corresponding experiments were simulated: experiments 3 and 4 (66 % RH, 29 % RH), experiments 7 and 11 (67 % RH, 26 % RH), experiments 8 and 13 (60 % RH, 25 % RH), and experiments 10 and 12 (50 % RH; 26 % RH). Equilibrium calculations were performed starting from both, low and high RH experiments. For experiments 5 and 14 performed at low RH values, the volatility distribution parameters could not be determined for the most volatile bin ($\alpha_5$). Therefore, equilibrium partitioning calculations were only performed starting from their corresponding experiments, namely experiment 1 (69 % RH), experiment 2 (67 % RH), and experiment 9 (56 % RH).

## 3 Experimental results

Figure 1 shows the suspended and wall-loss-corrected organic mass concentrations for all experiments listed in Table 1 as a function of OH exposure. SOA mass is rapidly formed and the wall-loss-corrected mass reaches a plateau at an approximate OH exposure of $2 \times 10^7 \, cm^{-3} \, h$. In the following, comparisons between the different experiments are carried out at an OH exposure of $2 \times 10^7 \, cm^{-3} \, h$. For illustrative purposes, the wall-loss-corrected aerosol yield is shown in Fig. S8, Supplement, as a function of α-pinene reacted to demonstrate its variability between different experimental conditions.

### 3.1 SOA yield dependence on RH, $NO_x$/α-pinene and aerosol seed composition

In Figure 2, we examine the relationship between the determined yields and the prevailing experimental conditions: RH, $NO_x$/α-pinene and seed composition. Lines connect experiments conducted under comparable conditions, but at different RH (dashed lines) and at different $NO_x$/α-pinene (solid lines). The visible effects of RH and $NO_x$ conditions on the yield in Figure 2 were statistically examined using a multilinear analysis (Fig. S9 and Table S3, Supplement). Three yield parameterizations were inter-compared and only the simplest model which represented significantly better the observations was considered for discussion.

The following features can be deduced from the analysis:



1/ SOA yields increase with $\alpha$-pinene concentrations, consistent with the semi-volatile nature of SOA compounds formed. We estimate that yields increase by approximately 2% when $\alpha$-pinene concentrations increase from 20 to 30 ppbv.

2/ A significant effect of the seed initial concentrations (or surface) on SOA yields was not observed. This may suggest (1) that SOA most likely forms its own phase and does not significantly partition into the seed aerosol (see below) and (2) that SOA condensation is not significantly affected by the vapor losses to the SC walls which can be diminished by increasing the aerosol surface. This is consistent with recent smog chamber results suggesting that for the $\alpha$-pinene system SOA formation is dominated by quasi-equilibrium growth and vapor losses to the walls do not depend on the seed concentrations, but rather on SOA (precursor) formation (oxidation) rates (Nah et al., 2016).

3/ SOA yields are significantly reduced under high $NO_x$ conditions (-3.3 % ± 0.6 %, p < 0.001), in agreement with literature data (Ng et al., 2007). Such a decrease indicates that SOA compounds formed under low $NO_x$ conditions are less volatile than those formed under high $NO_x$ conditions.

4/ We observed a clear influence of the RH on the yields, which increase on average by 1.5 ± 0.6% per 10% RH, for the range explored. This indicates that the particulate water content plays a central role in the partitioning of SOA compounds, either by altering the thermodynamic properties of the bulk phase (e.g. increasing the absorptive mass or decreasing the compound activity coefficients, non-reactive uptake) or by providing a reactive sink for semi-volatile species (e.g. formation of lower volatility compounds/oligomers in the bulk phase, reactive uptake). Additionally, the multilinear analysis suggests that the magnitude of the RH influence on SOA depends on the seed chemical nature (yields correlate with the interaction term between RH and seed composition), with a greater influence for the acidic seed (0.21 ± 0.03% per 1% RH, p < 0.001), the most hygroscopic aerosol, compared to the non-acidic seed (0.15 ± 0.03% per 1% RH, p < 0.001) and the hydrophobic seed (0.09 ± 0.04% per 1% RH, p = 0.05).

Overall, these results highlight the sensitivity of SOA yields to the prevailing oxidation conditions and the particle bulk-phase composition and water content and therefore the need for considering such conditions to obtain an accurate prediction of the SOA burden in the atmosphere. Nonetheless, results from this analysis should be regarded with some level of caution, owing to the limited size of the dataset. Despite the methodical assessment of the significance of the multilinear analysis results, we cannot unambiguously propose a mechanism by which water and seed hygroscopicity/acidity enhance SOA yields nor determine whether there is interplay between these two parameters. Nevertheless, we provide evidence that both of these parameters play a significant role in the formation or condensation of SOA species. In Sect. 4, using phase partitioning computation and AIOMFAC, we shall assess to which extent and under which conditions particulate water may alter the organic species' activity coefficients and as a consequence their absorptive partitioning.

## 3.2 SOA elemental composition

The effect of the experimental conditions on SOA chemical composition was investigated using the Van Krevelen space (Figure 3a). The overall region of the experimental data is very comparable for all experimental conditions and comparable to ambient data (Ng et al., 2011a). Data for experiments 1-14, with inorganic seeds, follow a similar slope with aging (-0.84): An increase in O:C during aerosol aging takes place under all





conditions. The O:C and H:C ratios of SOA produced with hydrophobic seed aerosol (expts. 15-17) show lower values than with inorganic seeds.

In Figure 3b, data are separated according to the wall-loss-corrected organic mass concentrations $C_{OA}$ (left panel: 2–6 µg m$^{-3}$; middle panel: 8–14 µg m$^{-3}$; right panel: 16–30 µg m$^{-3}$), to isolate the NO$_x$ and RH effects on the chemical composition from the possible influence of enhanced partitioning of semi-volatile organic species to the aerosol phase due to a higher SOA loading (Pfaffenberger et al., 2013). We observe that NO$_x$ levels have the highest influence on SOA elemental composition; namely, SOA formed at low NO$_x$ (marked with asterisks) is characterized by a higher H:C than that formed at high NO$_x$. This is even more pronounced at an early reaction stage (OH exposure $< 2 \times 10^6$ cm$^{-3}$ h, not shown in Figure 3), where fresh SOA products formed during all low NO$_x$ experiments have a higher H:C and lower O:C. Such influence of NO$_x$ levels on SOA elemental composition is consistent with our general understanding of gas-phase chemistry: Under low NO$_x$, substantial amounts of hydroperoxides and alcohols would result in a higher H:C ratio than e.g. carbonyls formed under high NO$_x$ conditions (Valorso et al., 2011). Also, as hydroperoxides and alcohols are substantially less volatile than carbonyls and less fragmentation is expected at lower NO$_x$ conditions, less oxygenated species with higher carbon number may partition to the particle phase at low NO$_x$, which would lead to lower O:C ratios.

NO$_x$ levels also affect the amount of organonitrate formed. Assuming that the entire nitrate signal arises from organonitrates (i.e. a maximum organonitrate contribution), we estimate molar ratios (Fig. S10, Supplement) of NO$_3$ to carbon of ~1:30 for high NO$_x$ and ~1:100 for low NO$_x$. Assuming oxidation product molecules with 10 carbon atoms, these ratios imply that every 3$^{rd}$ and every 10$^{th}$ molecule contains one NO$_3$ functional group, for high and low NO$_x$ conditions, respectively.

Conversely, we could not observe a significant effect of RH and seed composition on SOA elemental composition and degree of oxygenation, despite their significant influence on SOA yields. This may be due to the limited range of O:C ratio spanned by the different experiments in our case (on average $O:C \in [0.56 - 0.75]$ at an OH exposure of $2 \times 10^7$ cm$^{-3}$ h). Furthermore, we did not observe any significant dependence between these parameters and the ratio of organic fragments larger than $m/z$ 150 to the total organic mass, a proxy for oligomers measured with the HR-ToF-AMS, which would explain the observed increase in yields with the aerosol liquid water content (Fig. S11, Supplement).

### 3.3 Analysis of aerosol size-resolved chemical composition

As the seed composition and particulate water content appear to greatly influence SOA yields, we examine in this section the interaction between these parameters and SOA, through the investigation of the aerosol size-resolved chemical composition. This information is used later to infer the behavior of the absorptive organic phase and its mixing with the inorganic seed, while modeling the aerosol dynamics in the SC is beyond the scope of this study. Figure 4 shows the aerosol size resolved chemical composition at $0 \times 10^7$ cm$^{-3}$ h, $(0.5 \pm 0.2) \times 10^7$ cm$^{-3}$ h, $(1.0 \pm 0.3) \times 10^7$ cm$^{-3}$ h and $(2.0 \pm 0.5) \times 10^7$ cm$^{-3}$ h OH exposure for five experimental conditions (Data for additional experiments are available in the Supplement, Fig. S12). Figure S13 in the Supplement shows a 3-dimensional representation of the time dependent number and volume size distributions, measured by the SMPS.

For all experiments, the aerosol size distributions show two externally mixed aerosol populations, with a mode at lower diameters (~200 nm, mode 1) mostly containing SOA and another mode at higher diameters (~400 nm, mode 2) mostly consisting of the seed. We note that the particle size distribution evolved consistently under



different conditions, with the smallest seed particles growing with SOA condensation or coagulation (Fig. S12 and S13, Supplement). While we have detected an increase in particle number (Fig S14, Supplement), we note that intense nucleation events did not occur. For higher yields, the main SOA mass occurs in mode 1 and despite the sizeable increase of the yield with particulate water content (e.g. under acidic conditions and high RH), we did not observe a significant enhancement of SOA in mode 2. In addition, we did not note any statistically significant correlation between the initial seed volume and SOA yields; instead SOA growth seems to be driven by the favorable partitioning of semi-volatile species to smaller particles at an early stage of the experiment (Figure 4). Such behavior would imply that semi-volatile compounds do not additionally partition or react in the electrolyte rich phase on the timescale of this experiment, but rather the reactive or non-reactive uptake of these products onto the particles is enhanced with the increase of the initial particulate water content.

## 4 Phase partitioning calculation results

The organic yields and O:C ratios of the phase partitioning calculations are presented in Figure 7 and Figure 8, respectively, and listed in Table S9 of the Supplement. Each panel in Figure 7 compares organic yields for the cases org, id, sd, sdfr, and orgfr at the actual RH of the experiment (full color) and at the lower/higher RH of the corresponding experiment (light color). Table 2 gives the increase of organic yields from low to high RH of corresponding experiments as ratios [org yield (high RH) / org yield (low RH)].

The main objectives of the phase partitioning calculations are (1) estimating the impact of liquid water on SOA mixing properties (activity coefficients and liquid-liquid phase separation (LLPS)) and (2) determining the conditions or the potential model mixtures that can explain the observed yields and O:C ratios and their variation with RH. We note that model results are highly sensitive to the surrogates assumed, and the determination of SOA composition on a molecular level would considerably help confirming our results. Nevertheless, fitting both organic yields and O:C ratios significantly aid constraining the type of compounds that participate in partitioning (e.g. from a compound O:C ratio and vapor pressure its carbon number can be inferred). Accordingly, additional insights into the prevalent mechanisms by which the compounds form and evolve can be gained. For example, highly oxygenated compounds cannot be very volatile without significant fragmentation, whereas oligomerization leads to a significant decrease in the compounds' vapor pressure without necessarily increasing their O:C ratios.

### 4.1 Simulations with reported α-pinene photooxidation products (cases org, id, sd)

For case org, the contribution of model compounds ($\chi_{j,i}$) to the volatility bins at the actual RH could be optimized such that agreement was achieved between measured and calculated yields with deviations of less than 10 %, constituting a proof of concept of the applied approach. For most simulations, the $\chi_{j,i}$-values optimized for case org were also valid for cases id and sd.

In general, simulations assuming cases org, id and sd failed in predicting the change in SOA yields with RH (Figure 7) and led to a significant underestimation of the SOA O:C ratios (Figure 8). The only exception is the pair of experiments 10/12 performed at low $NO_x$ (1.9 ppbv) and high α-pinene levels (30.4 ppbv) for which the model could account for 87-99 % of the increase in SOA yield from low to high RH. This agreement was achieved although the model could not reproduce the high values of nor the change in the O:C ratios with RH





(the modelled O:C remained almost constant at 0.45-0.48 at low and high RH while the measured O:C increased from 0.56 at low RH to 0.64 at high RH). For the other experiments performed under low $NO_x$ conditions, model simulations accounted for only 43-75 % of the observed yield increase with RH. Simulated O:C values ranged from 0.48 to 0.55 and failed to reproduce the observed increase from 0.57 to 0.62 for experiments 8/13 and from

0.6 to 0.64 for experiments 7/11. Likewise, for high $NO_x$ conditions, model simulations for experiments 1/5, 2/5 and 9/14 could only account for 24-39 % of the observed yield increase from low to high RH and for 49-65 % for experiments 3/4. Simulated O:C values remained almost constant at 0.49-0.54 for all high $NO_x$ experiments and were considerably lower than the observed O:C ratios.

For case id the increased SOA yield at high RH, attributed to the additional partitioning of semi-volatile

compounds (norpinic acid, 2-hydroxyterpenylic acid, 10-oxopinonic acid and 4-oxopinonic acid) is a direct consequence of the increased absorptive mass due to the higher water content. For cases org and sd, partitioning to the condensed phase is reduced compared to the case id as AIOMFAC predicts activity coefficients greater than 1 for higher volatility compounds, e.g. 10-oxopinonic acid and 4-oxopinonic acid. This effect is even enhanced for case sd when partitioning to the total condensed phase including the seed aerosol is simulated.

Although LLPS is predicted for all simulations, the salting-out effect of AS which partitions to some degree also to the organic phase leads to a further decrease of the organic yield at high RH compared to the case id.

In summary our results suggest that with the reported model compounds for α-pinene photooxidation, the measured O:C and the increase of organic yields with RH cannot be simulated. Therefore, we explored whether the formation of fragmented and more oxidized products may explain the high O:C ratio observed and the high

sensitivity of the yields to RH.

### 4.2    Simulations including fragmented products (cases sdfr and orgfr)

Non-fragmented products (e.g. highly oxygenated C10 or dimers) would be low-volatility (LVOC) or extremely low-volatility organic compounds (ELVOC) (Zuend and Seinfeld, 2012; Donahue et al., 2006; 2012b) with effective saturation concentrations $C^* \leq 0.1$ µg m$^{-3}$, when their O:C ratio is as high as the observed O:C ratio.

Therefore, they are expected to be in the particle phase independent of the prevalent RH and the inclusion in the model of additional amounts of these products would not help explaining the observed difference in the yields at different RH, leading to an overestimation of OA mass at low RH. This implies that to better capture the measured O:C range without overestimating SOA yields, fragmented products with shorter backbones need to be introduced, in accordance with Donahue et al. (2012a).

Low molecular weight compounds resulting from fragmentation were therefore added to the volatility bins with $C^* = 1$-100 µg m$^{-3}$, namely, 3-oxoadipic acid to the volatility bin with $C^* = 1$ µg m$^{-3}$, glutaric acid to the volatility bin with $C^* = 10$ µg m$^{-3}$ and 5-COOH-3-OH-pentanal and succinic acid to $C^* = 100$ µg m$^{-3}$. For case sdfr $\alpha_i$-parameters were optimized assuming equilibrium partitioning of SOA to the whole condensed phase including the seed aerosol. For case orgfr absorption to the organic phase only was assumed. While the organic

and the inorganic phases seem to form externally mixed particles based on the chemically resolved size distribution, examining both cases provides valuable insights into the impact of the presence of an inorganic aerosol seed in the system. Tables S7 and S8 of the Supplement provide the contribution of model compounds ($\chi_{j,i}$ values) to volatility bins $C_i^*$ for these cases. The organic yields and O:C ratios for cases sdfr and orgfr are shown in Figure 7 and Figure 8, respectively. In Table 2 the relative increase in SOA yields from low to high RH





is given. Cases sdfr and orgfr are similarly successful in reproducing SOA yields and O:C ratios. However, the variability in the modelled yields when using different volatility distributions for cases sdfr and orgfr is larger than for cases org, id, and sd (Figure 7).

For experiments carried out at low $NO_x$ conditions (experiments 7/11, 8/13 with $NO_x$ < 2 ppbv), both cases explained the observed increase in the organic yields with RH and improved the agreement between modelled and observed O:C ratios. For experiments 8/13, both cases accurately captured the increase in SOA yields from low to high RH with deviations < 2 %. Modelled O:C agreed very well with measurements for case sdfr with deviations ≤ 1% and slightly worse for case orgfr with deviations of ≤ 5%. For experiments 7/11, cases sdfr and orgfr reproduced satisfactorily (deviations ≤ 11%) the measured yields for calculations with the volatility distributions determined for experiment 11, but overestimated the observed increase of O:C from low (O:C = 0.60) to high RH (O:C = 0.64). Using the volatility distributions determined for experiment 7, the low yields at low RH could not be achieved. For experiments 10/12, cases org, id, and sd were able to simulate the increase of organic yield with the oxidation products reported in literature but failed to reproduce the observed O:C ratios. Cases sdfr and orgfr with additional oxidized and fragmented products showed improved agreement of observed yields and reproduced successfully the observed O:C ratios.

Under high $NO_x$ conditions, cases sdfr and orgfr could only simulate satisfactorily SOA yields and O:C ratios observed during experiments 3/4. For experiments 1-2/5 and 9/14, model and measurement agreement was less satisfactory. The model could not reproduce the low yields at low RH observed during experiments 5 and 14 and hence the strong reduction of the yields observed with the decrease in RH is underestimated (the simulations predict only 52-66 % of the observed change). Additionally, for these experiments the model underestimates the O:C ratios (average measured O:C = 0.66 and 0.68, at low and high RH, respectively). In the following, we examine and discuss the potential reasons that might explain model-measurements disagreements.

### 4.3 Simulations including organonitrates

We examined whether the discrepancy between modelled and measured yields may be ascribable to the selection of the model compounds at high $NO_x$, by introducing additional surrogates - specifically organonitrates - expected to be representative of the compounds formed under these conditions. Assuming that the $NO_3$ signal in the HR-ToF-AMS originates from organonitrates, every 3[rd] molecule should contain one $ONO_2$ functional group at high $NO_x$ conditions. We tested whether adding organonitrates described by Valorso et al. (2011) would improve the agreement between measurements and observations for the high $NO_x$ experiments. However, this was not the case. Such sensitivity tests could not be performed for cases including partitioning to the seed aerosol because interaction parameters between organonitrates and sulfate are not available in AIOMFAC.

### 4.4 Simulations including higher volatility oxidation products

The Monte Carlo simulations enabled the determination of $\alpha_i$-parameters for volatility bins $C* = 0.1$-100 µg m$^{-3}$. $\alpha_i$-Parameters in the volatility bin with $C* = 1000$ µg m$^{-3}$ could not be reliably extracted. Nevertheless, SOA products belonging to this bin are present in the chamber and could partition to some extent to the condensed phase. We investigated whether the presence of substances with an effective saturation concentration of 1000 µg m$^{-3}$ could reproduce the low SOA yields at low RH and its strong increase at high RH observed for the high $NO_x$ experiments 1/5, 2/5, and 9/14. This was achieved by replacing a part of the substances in the volatility bin





$C* = 100$ µg m$^{-3}$ by pinalic acid, terpenylic acid and 3-2-oxopropanylolxypropanoic acid (Table S4, Supplement). While the addition of these compounds in the model could reproduce the low yields at low RH, the yields at high RH were strongly underestimated. Accordingly, the introduction of organonitrates or higher volatility compounds in the model did not improve the agreement between modelled and observed yields and

O:C ratios, suggesting that equilibrium partitioning alone cannot explain the strong SOA yield increase under high NO$_x$ conditions.

### 4.5      Partitioning of individual components to the gas phase and condensed phases

The phase partitioning calculations do not only allow the simulation of the total organic yield and average O:C ratio of the condensed products, but also provide insights to the partitioning of individual compounds to the gas

phase and the condensed phases. Phase partitioning of individual model compounds into the gas and condensed phases for case sdfr is examined in Figure 9 for experiments 8/13 and in Figure 10 for experiments 3/4. Detailed results of the phase partitioning calculations are listed in Tables S10 and S11 in the Supplement. For experiments 8/13, the model predicts an LLPS into an organic-rich phase (op) and a predominantly electrolyte-like phase (ep). Overall, organic compounds are predominantly in the organic phase at both RH with ep/op $< 1 \cdot 10^{-5}$ at 25 %

RH, and ep/op $\approx 0.003-0.008$ at 60 % RH. Compounds in volatility bins $C* = 0.01$ and 0.1 µg m$^{-3}$ are mainly present in the condensed phases, while compounds in volatility bin $C* = 100$ µg m$^{-3}$ show preferred partitioning to the gas phase. The strongest increase in the condensed phase when RH is increased from 25 % to 60 % is observed for the model compounds assigned to $C* = 100$ µg m$^{-3}$. Moderately oxygenated species (4-oxopinonic acid and 10-oxopinonic acid) in this bin show a moderate increase (about a factor of two) driven by the increase

of the absorptive mass. This increased partitioning is limited by an increase in the activity coefficients of these compounds (for experiment 8 from 1.69 and 1.63 at low RH to 2.70 and 2.93 at high RH and for experiment 13 from 1.49 and 1.54 at low RH to 2.88 and 3.24 at high RH). Conversely, the 5-fold enhanced partitioning of the fragmented and more functionalized compounds (5-COOH-3-OH-pentanal and succinic acid) into the condensed phase at high RH is driven by the increase of the absorptive mass and the slight decrease of the compounds'

activity coefficients (for experiment 8 from 0.84 and 0.51 at low RH to 0.68 and 0.43 at high RH and for experiment 13 from 0.90 and 0.56 at low RH to 0.69 and 0.44 at high RH).

For experiments 3/4, model results obtained from the parameterization of the yields in experiment 4 are highly sensitive to the assumed volatility distribution. As shown in Figure 10, when introducing volatility distributions characterized by high contributions of semi-volatile oxygenated compounds pertaining to the volatility bin $C* = $

100 µg m$^{-3}$, the model predicts a liquid phase mixing. Note that when mixing is predicted, the model tends to overestimate the O:C ratio (O:C = 0.81 at high RH for the volatility distribution #2 shown in the upper panels of Figure 10) and the yields at both high and low RH. Conversely, when volatility distributions with less volatile and less oxygenated compounds are used (e.g. volatility distribution #93), lower yields and lower O:C ratios (0.69 for volatility distribution #93 shown in the lower panels of Figure 10) are obtained and LLPS is predicted.

For such O:C ratios LLPS has also been observed for experiments performed with model mixtures (e.g. Song et al., 2012). For experiment 4, the volume of the electrolyte phase is larger than that of the organic-rich phase and there is considerable partitioning of organic compounds to the electrolyte phase (ep/op = 2.26). These observations illustrate that for compounds with high O:C ratios, small differences in the volatility distribution parameters can lead to totally different phase partitioning. This highlights the suitability of the approach used,

enabling the assessment of the modelled results to uncertainties in the volatility distribution determination.





Irrespective of these differences, none of the volatility distributions used could reproduce the measured yields at both high and low RH, likely due to the deficient representation in the model of the interactions between the acidic aerosol and the organic compounds - and of the chemical processes occurring under acidic/high $NO_x$ conditions.

## 4.6      Modeling considerations and limitations

Model simulations were carried out by considering two $\alpha$-pinene C8-C10 photo-oxidation products per volatility bin for cases org, id and sd. This number was sufficient because the hydrophilicity of these compounds with consistently high carbon number correlates with their volatility. Therefore the sensitivity of model predictions to the contribution of these compounds to the volatility bins (described by $\chi_{j,i}$ values) is relatively low. This model

setting, corresponding to a pseudo-one-dimensional VBS where the compounds' vapor pressure and degree of oxidation correlate, was proven to be insufficient for an accurate description of both SOA mass and degree of oxygenation. By contrast, for cases sdfr and orgfr, which include more oxidized short chain products, hydrophilicity and volatility may be varied more independently, which introduces an additional degree of freedom and would correspond to a pseudo-two-dimensional VBS (2D-VBS), as described by Donahue et al.

(2011). While we show that model predictions based on this setting are highly dependent on the volatility distribution parameters and model compounds assumed, in general the introduction of fragmented more oxidized compounds reproduced well the high observed O:C ratios and the increase of O:C and SOA yields with RH, specifically at low $NO_x$. The estimated volatility distributions and average carbon (C~6) and oxygen (O~4) numbers when considering fragmented products are in agreement with chemical speciation analysis previously

reported for the same system (Chhabra et al., 2015). The analysis shows that for such semi-volatile products with O:C ratios in the order of ~0.6 an increase of RH from 23-29 % to 60-69 % induces a mass increase by up to a factor of three, driven by the higher particle water content and the lower activity coefficients of the more fragmented products at high RH.

The measured O:C ratio is a key parameter for constraining the model. We do not expect the accuracy of the O:C

ratios determined by the HR-ToF-AMS to be less than 20-30% (Aiken et al., 2007; Canagaratna et al., 2015; Pieber et al.). Here, we have used the high resolution parameterization proposed by Aiken et al. (2007), while a more recent parameterization by Canagaratna et al. (2015) is now available. The utilization of the latter would result in even higher O:C ratios (by 20%), which would require increasing even further the degree of oxidation of the fragmented compounds and imply that the model would predict even a higher sensitivity of the yields to

RH.

Under high $NO_x$ (and low pH), the model could not reproduce the factor of six increase in yields at high RH, using a variety of chemically dissimilar surrogate compounds. This indicates that the increased absorptive uptake of these compounds due to the increase in SOA mass and the decrease in the compounds' activity coefficients cannot explain the observed enhancements alone. Under these conditions, additional processes may likely play a

role in the enhancement of OA with RH. $NO_x$ concentrations have a dramatic influence on SOA chemical composition and volatility. Therefore, the discrepancy between model and measured yields at high $NO_x$ conditions may be explained by either an inadequate representation of SOA surrogates in the model and their interaction parameters with the seed or by an enhanced reactive uptake of SOA species formed under high $NO_x$ conditions, such as carbonyls that are reported to instigate the formation of lower volatility compounds in the



particle phase (Shiraiwa et al., 2013 and references therein). Given the dearth of additional chemically resolved measurements of the particle phase species formed under different conditions, the mechanism by which RH enhances the uptake of SOA species under high NO$_x$ (low pH) remains currently undetermined.

Cases org, id, and orgfr assumed gas-particle partitioning into the organic aerosol only. In this case, the role of
the seed aerosol is restricted to providing a substrate for nucleation of organic vapors. This is the case for effloresced seed particles. For liquid seed aerosols this is equivalent to a complete organic/electrolyte phase separation with no partitioning of inorganic ions to the organic phase. For cases sd and sdfr equilibrium partitioning between the gas phase and the entire condensed phase including the seed aerosol is assumed leading in most cases to LLPS. The current model does not yet contain interaction parameters of bisulfate with all
involved organic functional groups. Therefore, ammonium bisulfate was treated as ammonium sulfate in the model, and we are not capable of distinguishing whether the enhanced partitioning of semi-volatile vapors in the acidic medium is attributable to additional reactions in the bulk phase (catalysed at lower pH) or to an enhanced solubility of SOA species.

Considering the HR-ToF-AMS data in Figure 4 discussed in Sect. 3.3, LLPS manifests in the presence of two
externally mixed particle populations. The formation of these two populations may occur by the homogeneous or heterogeneous nucleation of highly oxidized non-volatile products. Homogeneous nucleation implies new particle formation, while heterogeneous nucleation proceeds via condensational growth. Both processes are expected to create small organic rich particles, providing an organic absorptive phase into which additional semi-volatile compounds may preferentially partition. When the organic and electrolyte phases are present in different
particles the two phases communicate via gas phase diffusion, and equilibration occurs depending on the volatility of the components. For compounds with $C_j^* = 0.1\text{-}100\ \mu g\ m^{-3}$ equilibration occurs within time-scales of minutes to tens of minutes, assuming no bulk phase diffusion limitations (Marcolli et al., 2004). It is expected that in the larger particle mode a liquid-liquid phase separation will not establish because the inorganic ions would exert a salting-out effect driving the organic compounds to partition to the gas-phase or into the smaller
organic-rich particles, which would deplete even further these particles from the organic material. Under such scenario an externally-mixed phase-separated aerosol might evolve in the smog chamber.

The scenario outlined above is based on equilibrium partitioning and does not invoke diffusion limitations within the condensed phase. Recent evidence may challenge this assumption, suggesting that SOA may adopt a highly-viscous state (e.g. Virtanen et al., 2010; Koop et al., 2011), where bulk diffusion and evaporation are kinetically
limited. However, while such behavior occurs under certain conditions, e.g. low temperature and low relative humidity, we expect that this is not the case for the aerosol investigated in this study. Saleh et al. (2013) showed that SOA from α-pinene ozonolysis reaches equilibrium with the gas phase within tens of minutes at low mass loadings (2-12 $\mu g\ m^{-3}$) upon a step-change in temperature. Robinson et al. (2013) determined by aerosol mixing experiments that the diffusion coefficient in α-pinene-derived SOA is high enough for mixing on a time scale of
minutes. Fast mixing is further supported by measurements of ambient OA (Yatavelli et al., 2014) showing that biogenic SOA reaches equilibrium within atmospheric time scales, under similar conditions as those in our chamber. Therefore, we expect that, if thermodynamically favorable, liquid-liquid mixing would have occurred under the time-scales of our experiments and a unimodal particle population would have emerged. However, consistent with model predictions, this is not the case.



## 5    Summary and conclusions

We conducted a series of smog chamber experiments to investigate the impact of $NO_x$/VOC ratios, RH and the seed aerosol composition on the yield and the degree of oxygenation of SOA produced from α-pinene photooxidation. We developed a novel approach based on Monte Carlo simulations to determine SOA volatility

distributions from the measured yields using the volatility basis set framework. Measured yields spanned an order of magnitude (2-20 %), depending on the prevailing oxidation conditions, the particle bulk-phase composition and the water content. This underlines the need of considering these conditions for an accurate prediction of the SOA burden in the atmosphere. The yields increased dramatically with RH, aerosol acidity and at low $NO_x$. The aerosol bulk chemical composition measured by the HR-ToF-AMS appears to be mostly

dependent on $NO_x$/VOC ratios.

We investigated whether equilibrium partitioning between the gas and the condensed phase(s) can reproduce the measured SOA yield and O:C ratio at low and high RH. For this, the thermodynamic group-contribution model AIOMFAC was used to examine the partitioning of a range of selected surrogate compounds with different volatilities and O:C ratios as a function of their activity coefficients and the particle water content; properties that

are altered with the variation of RH. In practice, two to four surrogate compounds were assigned to each volatility bin based on the volatility distributions derived from experimental data and the compounds' effective saturation concentration, which depends on the pure component saturation vapor pressures. The RH was then varied in the model and changes in SOA mass and O:C ratios were monitored. There are large discrepancies between vapor pressures determined for semi-volatile and low volatility compounds depending on the

measurement techniques (Huisman et al., 2013; Bilde et al., 2015), which introduce inaccuracies in saturation vapor pressure estimations. However, these uncertainties do not affect the approach applied in this study because an incorrect assignment of a model compound to a volatility bin does not change the experimentally derived volatility bin distribution. The share of the individual model compounds to the volatility bins was used as fitting parameter to achieve agreement between measurements and calculations. In addition, as a set of volatility

distribution functions was obtained for each experiment through the Monte Carlo simulations, the approach is proven very effective in assessing the sensitivity of equilibrium partitioning calculations on the volatility distribution input parameters.

Modelling results show that in order to simultaneously fit the SOA masses and O:C ratios, compounds arising from fragmentation need to be considered. Under these conditions, the model predicts that an increase in RH

from ~25% to ~60% may lead to a three-fold enhancement in SOA mass, due to the increase in the absorptive mass and the slight decrease in the compounds' activity coefficients. While the magnitude of this increase is consistent with the experimental observation at low $NO_x$, equilibrium partitioning alone could not explain the strong increase in the yields with RH observed for the high $NO_x$ experiments (factor ~6). This suggests that other processes including the reactive uptake of semi-volatile species into the liquid phase may occur and be enhanced

at higher RH, especially for compounds formed under high $NO_x$ conditions such as carbonyls. Future studies should investigate the dependence of SOA compounds on RH, $NO_x$ and aerosol acidity at a molecular level. Such measurements shall provide additional insights into the chemical nature of the compounds that additionally partition or react at higher RH and also the mechanisms via which such processes occur.

For most of the cases studied, the model predicts liquid-liquid phase separation into an organic and an electrolyte

phase. Considering the size-resolved particle chemical composition, this phase separation is likely not realized



within single particles but the aerosol population splits up into a predominantly organic mode at ~200 nm and a predominantly inorganic mode at ~400 nm. Such liquid-liquid phase separation is expected to occur under most ambient conditions, i.e. similar levels of OA and sulfate, O:C < 0.8, and RH < 70–80 %.

5 *Acknowledgements.* This work has been supported by the EU 7th Framework projects EUROCHAMP-2 and PEGASOS, as well as the Swiss National Science Foundation (Ambizione PZ00P2_131673, SAPMAV 200021_13016), the EU commission (FP7, COFUND: PSI-Fellow, grant agreement n.° 290605) and the joint CCES-CCEM project OPTIWARES. We thank René Richter and Günther Wehrle for their technical support at the smog chamber, and in addition Michel J. Rossi, Martin Gysel, Neil Donahue and Barbara Turpin for the
10 helpful discussions. We thank Andreas Zuend for providing the Fortran code for phase partitioning calculations and AIOMFAC and helpful discussions.





## 6    Tables and Figures

Table 1. Overview of experimental conditions. Seed types {AHS = ammonium hydrogen sulfate (NH$_4$HSO$_4$); SA = sulfuric acid (H$_2$SO$_4$); AS = ammonium sulfate ((NH$_4$)$_2$SO$_4$); CF = fluorinated carbon (see Sect. 2.2)} and their assumed phase states: (L) = liquid; (-) = liquid and/or solid. Initial seed mass concentrations; relative humidity (RH);

5    measured mean NO$_x$ concentrations during low NO$_x$ experiments (marked with an asterisk*) and measured initial NO$_x$ concentrations during high NO$_x$ experiments; measured initial α-pinene concentrations (which reacted before an OH exposure of $(2.0 \pm 0.5) \times 10^7$ cm$^{-3}$ h) and their estimated fractions reacted with OH radicals in %, rest reacted with O$_3$). NO$_x$/α-pinene ratios; wall-loss corrected organic mass concentrations ($C_{OA}$)and corresponding yields $Y$ averaged over the OH exposure of $(2.0 \pm 0.5) \times 10^7$ cm$^{-3}$ h. Standard deviations (1sd) given in brackets are measurement

10    variability. Horizontal lines separate experiments with different i) seed composition, ii) RH, iii) NO$_x$/α-pinene. Blank experiments (B1, B2 and B3) are listed at the very bottom.

| No | seed type (phase) | initial | RH | NO$_x$ initial or mean(*) | α-pin initial = reacted | α-pin decay by OH | NO$_x$/ α-pin | $C_{OA}$ (wlc) at OH exposure: $(2.0\pm0.5) \times 10^7$ cm$^{-3}$ h | Yield, $Y$ (wlc) |
|---|---|---|---|---|---|---|---|---|---|
| | | µg m$^{-3}$ | % | ppbv | ppbv | % | | µg m$^{-3}$ | |
| 1 | AHS+SA (L) | 8.0(0.5) | 69(2) | 44.4(0.8) | 20.7 | 80.0 | 2.1 | 13.4(0.2) | 0.115 |
| 2 | AHS+SA (L) | 12.3(0.5) | 67(2) | 70.4(1.3) | 18.7 | 80.3 | 3.8 | 8.6(0.1) | 0.081 |
| 3 | AHS+SA (L) | 4.9(0.3) | 66(2) | 19.6(0.7) | 16.1 | 81.6 | 1.2 | 12.6(0.6) | 0.138 |
| 4 | AHS+SA (L) | 4.7(0.2) | 29(1) | 23.6(0.6) | 19.9 | 81.3 | 1.2 | 3.9(0.0) | 0.035 |
| 5 | AHS+SA (L) | 8.0(0.3) | 28(1) | 52.1(0.6) | 20.3 | 74.6 | 2.6 | 2.1(0.1) | 0.018 |
| 6 | AHS+SA (L) | 5.2(0.3) | 27(1) | 1.3(0.4)* | 18.3 | 87.8 | 0.071 | 12.0(0.5) | 0.116 |
| 7 | AS+AHS (L) | 8.8(0.4) | 67(1) | 0.7(0.2)* | 20 | 81.9 | 0.037 | 16.2(0.4) | 0.143 |
| 8 | AS+AHS (L) | 4.3(0.6) | 60(1) | 1.0(0.2)* | 18.7 | 86.5 | 0.052 | 12.3(0.4) | 0.116 |
| 9 | AS+AHS (L) | 5.5(0.2) | 56(2) | 65.8(0.8) | 30.9 | 65.0 | 2.1 | 11.1(0.1) | 0.064 |
| 10 | AS+AHS (L) | 4.4(0.2) | 50(1) | 1.3(0.2)* | 30.6 | 79.2 | 0.041 | 29.6(1.1) | 0.171 |
| 11 | AS+AHS (-) | 4.1(0.2) | 26(1) | 0.7(0.3)* | 18.9 | 81.3 | 0.039 | 5.5(0.2) | 0.051 |
| 12 | AS+AHS (-) | 3.6(0.1) | 26(1) | 1.9(0.4)* | 30.5 | 78.4 | 0.062 | 20.3(1.3) | 0.118 |
| 13 | AS+AHS (-) | 8.2(0.3) | 25(1) | 1.1(0.3)* | 19.6 | 87.4 | 0.055 | 5.3(0.1) | 0.048 |
| 14 | AS+AHS (-) | 3.2(0.2) | 23(1) | 75.1(0.7) | 27.8 | 69.4 | 2.7 | 2.4(0.1) | 0.015 |
| 15 | CF (L) | 7.1(0.3) | 58(1) | 56.2(0.7) | 31.7 | 67.9 | 1.8 | 9.8(0.1) | 0.054 |
| 16 | CF (L) | 10.0(0.5) | 58(2) | 58.3(0.5) | 31.3 | 73.9 | 1.9 | 10.7(0.1) | 0.061 |
| 17 | CF (L) | 6.7(0.1) | 26(1) | 53.3(0.6) | 30.5 | 69.7 | 1.7 | 4.7(0.1) | 0.021 |
| B1 | CF (L) | 0.3(0.1) | 58(2) | 53.0(0.6) | - | - | - | < 0.1 | - |
| B2 | AS+AHS (L) | 4.5(0.7) | 68(2) | 0.9(0.5)* | - | - | - | 2.8(1.0) | - |
| B3 | AHS+SA (L) | 3.8(0.2) | 75(3) | 1.4(0.3)* | - | - | - | 0.5(0.1) | - |





**Table 2. Increase of organic yield from low to high RH for the different experiments as org yield (high RH) /org yield (low RH). The second column lists the ratio of measured organic yields enhancements with RH. In the five last columns the ratios of organic yields calculated at high and low RH for the different experiments are given (average value of the ten volatility distributions with lowest and highest values in brackets in the second row).**

| Experiments | | Calculations | | | | | |
|---|---|---|---|---|---|---|---|
| Expt. No. | yield ratio | Exp | Case org | Case id | Case sd | Case sdfr | Case orgfr |
| 1 / 5 | 6.38 | 1 | 1.62 (1.48-1.86) | 2.02 (1.67-2.57) | 1.54 (1.37-1.77) | 4.23 (2.50-6.70) | 4.34 (2.89-6.62) |
| 2 / 5 | 4.10 | 2 | 1.42 (1.23-1.47) | 1.61 (1.31-1.73) | 1.31 (1.18-1.35) | 2.74 (1.33-6.24) | 2.24 (1.35-2.92) |
| 3 / 4 | 3.23 | 3 | 1.64 (1.44-1.86) | 2.07 (1.88-2.38) | 1.57 (1.48-1.73) | 3.67 (2.74-5.29) | 3.13 (2.53-4.14) |
| 3 / 4 | 3.23 | 4 | 1.76 (1.58-1.90) | 2.09 (1.78-2.41) | 1.58 (1.44-1.71) | 2.93 (1.91-4.88) | 3.12 (2.13-4.09) |
| 7 / 11 | 2.95 | 7 | 1.35 (1.25-1.47) | 1.52 (1.35-1.80) | 1.28 (1.20-1.39) | 2.16 (1.48-3.98) | 2.38 (1.64-3.69) |
| 8 / 13 | 2.32 | 8 | 1.43 (1.30-1.58) | 1.61 (1.43-1.87) | 1.36 (1.27-1.50) | 2.66 (1.60-4.79) | 2.44 (1.73-3.57) |
| 9 / 14 | 4.63 | 9 | 1.43 (1.32-1.59) | 1.56 (1.38-1.82) | 1.33 (1.23-1.48) | 2.45 (1.61-4.70) | 2.69 (1.75-4.24) |
| 10 / 12 | 1.46 | 10 | 1.29 (1.13-1.52) | 1.38 (1.22-1.67) | 1.28 (1.17-1.49) | 1.44 (1.18-1.66) | 1.55 (1.26-2.17) |
| 7 / 11 | 2.95 | 11 | 1.80 (1.52-2.21) | 2.22 (1.69-3.09) | 1.67 (1.40-2.05) | 2.74 (1.75-4.03) | 3.38 (1.87-5.61) |
| 10 / 12 | 1.46 | 12 | 1.34 (1.21-1.48) | 1.48 (1.27-1.71) | 1.33 (1.19-1.49) | 1.34 (1.19-1.65) | 1.70 (1.33-2.06) |
| 8 / 13 | 2.32 | 13 | 1.54 (1.39-1.68) | 1.74 (1.52-1.91) | 1.41 (1.30-1.50) | 2.73 (1.29-6.06) | 2.54 (1.39-4.90) |





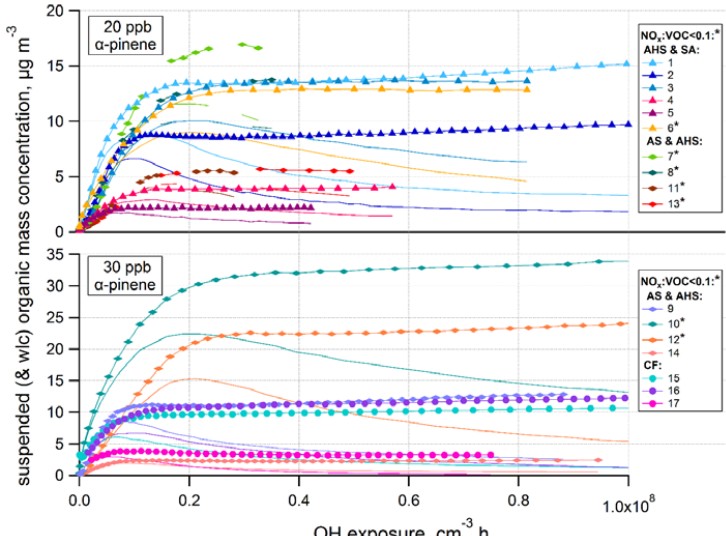

**Figure 1.** 20-min-averaged wall-loss-corrected (symbols & lines) and suspended (lines) organic mass concentrations as a function of OH exposure. Data is separated according to similar initial α-pinene concentration (20 ppbv - top panel, 30 ppbv - bottom panel). Experiment numbers are given in the legend and classified by seed composition; asterisks indicate low $NO_x$ experiments.



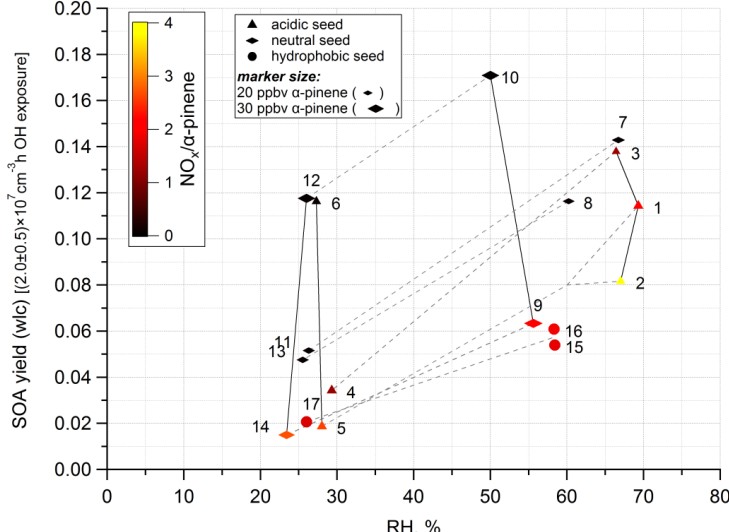

**Figure 2.** Average wall-loss-corrected yields $Y$ at $(2.0 \pm 0.5) \times 10^7$ cm$^{-3}$ h OH exposure as a function of RH. Symbol sizes represent α-pinene reacted, symbol colors represent NO$_x$/α-pinene and symbol shapes represent seed composition according to Table 1. Experiments with similar NO$_x$/α-pinene, seed composition and α-pinene reacted are connected with dashed lines showing the increase in yield for increased RH. Experiment 6 has no counterpart experiment at high RH. Experiments with similar RH, α-pinene reacted and seed composition are connected with solid lines showing the increased yield with decreasing NO$_x$/α-pinene ratio.





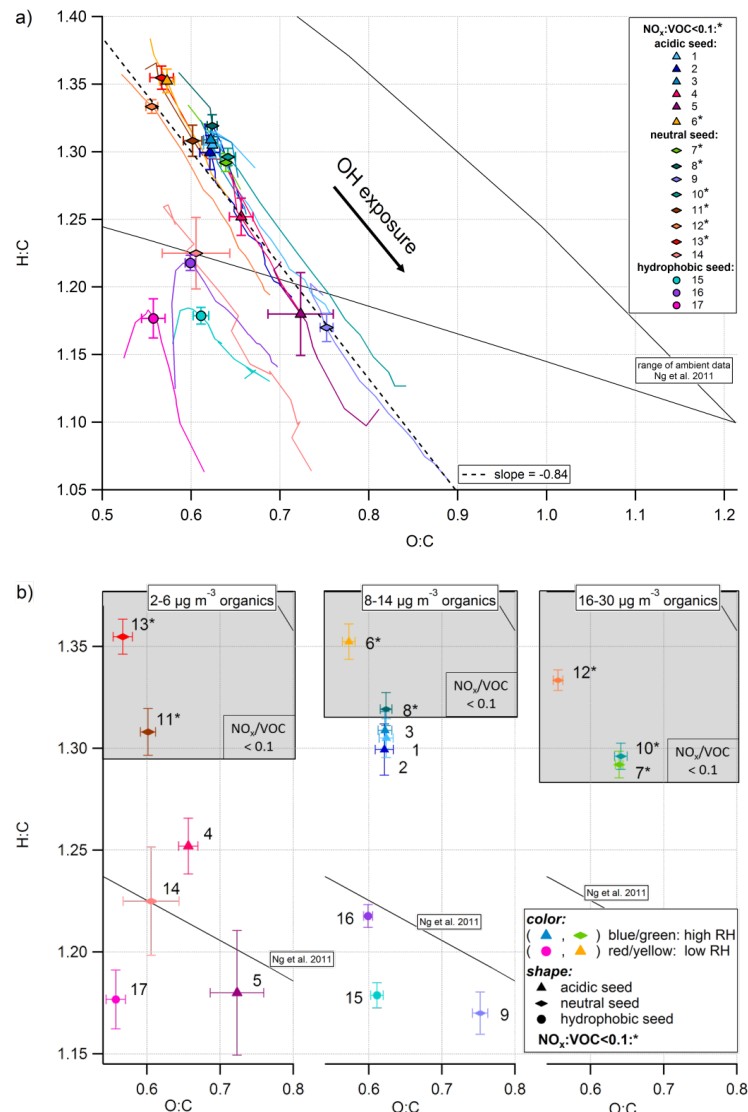

**Figure 3. Van Krevelen diagrams: Mean (and 1 standard deviation measurement variability) H:C versus O:C at OH**
5 **exposure $(2.0 \pm 0.5) \times 10^7$ cm$^{-3}$ h (symbols). Symbol colors indicate RH, symbol shapes the seed composition and the asterisk low NO$_x$ experiments. (a) The dashed line represents the (least orthogonal distance) fitted slope of experiments (1-14) with inorganic seed: -0.84. The triangular shaped solid lines) represents the range of ambient SOA (Ng et al., 2011a). 1 h-averages of H:C versus O:C (for OH exposure > $2 \times 10^6$ cm$^{-3}$ h) are given by the colored lines. Experiments 15-17 show the influence of organic seed compounds (data shown from suspended organic mass > 0.3**
10 **µg m$^{-3}$). (b) The data was split in three groups according to their wall-loss-corrected organic mass concentrations to exclude concentration effects (left panel: 2-6 µg m$^{-3}$; middle panel: 8-14 µg m$^{-3}$; right panel: 16-30 µg m$^{-3}$). The grey shaded areas include all low NO$_x$ experiments.**







**Figure 4.** Evolution of *size distributions* from the **HR-ToF-AMS**. Measured organic, SO₄, NH₄, NO₃ mass distributions for OH exposures of $0\times$, $(0.5 \pm 0.2)\times$, $(1.0 \pm 0.3)\times$ and $(2.0 \pm 0.5)\times 10^7$ cm⁻³ h. Black lines represent estimated liquid water content (method: Sect. 2.3; RH and individual *GF*s given in the legend, percentage in brackets: fractions of SO₄). The calculated dry and wet surface distributions are shown as dashed lines on the right axes. (a) High NO$_x$ with more acidic seed [H₂SO₄ & NH₄HSO₄]: Experiments 3 and 4 with NO$_x$/α-pinene = 1.2 and experiment 2 with NO$_x$/α-pinene = 3.8. (b) Low NO$_x$ with less acidic seed [(NH₄)₂SO₄ & NH₄HSO₄]: Experiments 8 and 13 with NO$_x$/α-pinene ≈ 0.05. Additional figures in the Supplement (Fig. S12).





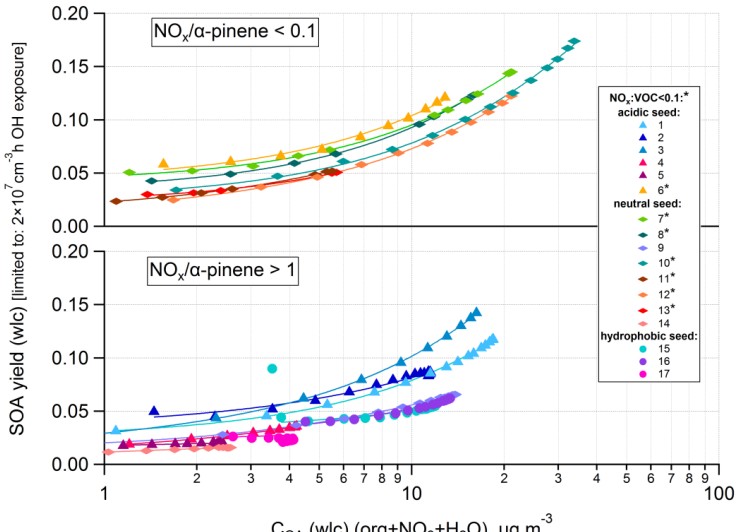

**Figure 5. Measured (symbols) and parameterized (lines) wall-loss-corrected SOA yield, *Y*, as a function of wall-loss-corrected absorptive mass concentration (*C_OA*: organics + NO_3 + H_2O) for low NO_x experiments (upper panel) and high NO_x experiments (lower panel). Data was limited to an OH exposure of $2 \times 10^7$ cm$^{-3}$ h and averaged over 20 minutes. Symbol colors indicate the RH, symbol shapes the seed composition and the asterisks the NO_x/VOC ratio.**

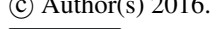


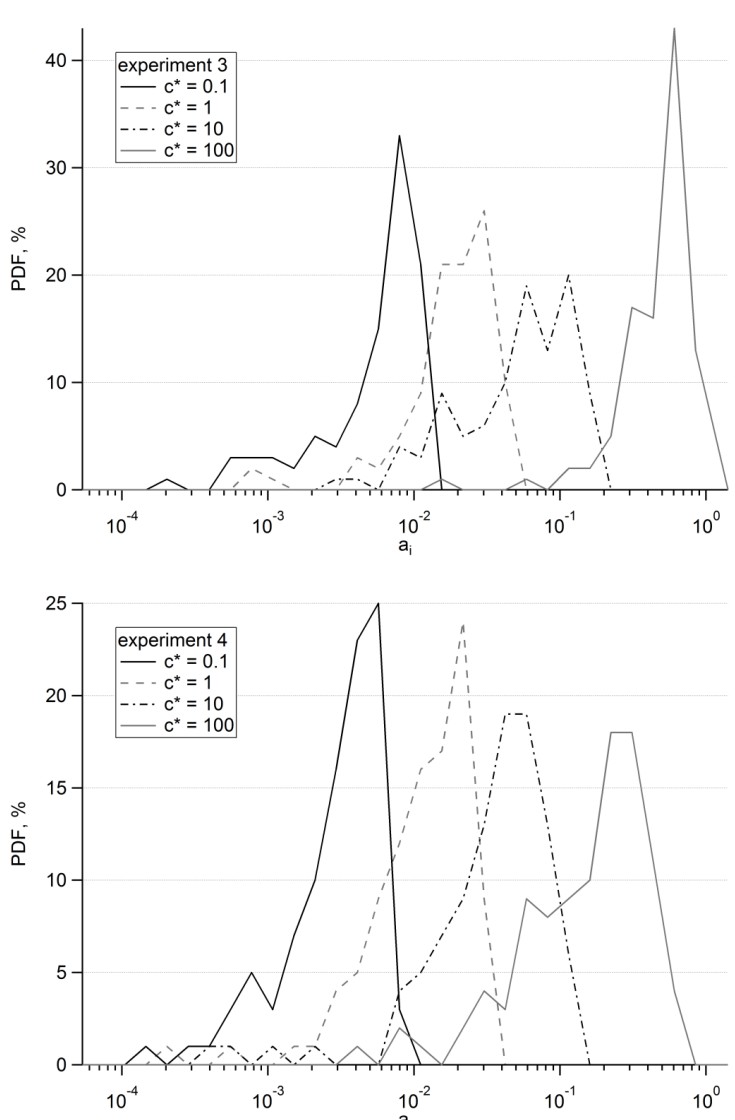

**Figure 6.** Probability density functions (PDF) of $\alpha_i$'s for volatility bins ($C_i^* = 0.01$, 0.1, 1, 10 and 100 µg m$^{-3}$) for one high *RH* (exp. 3, upper panel) experiment and one low *RH* (exp. 4, lower panel).



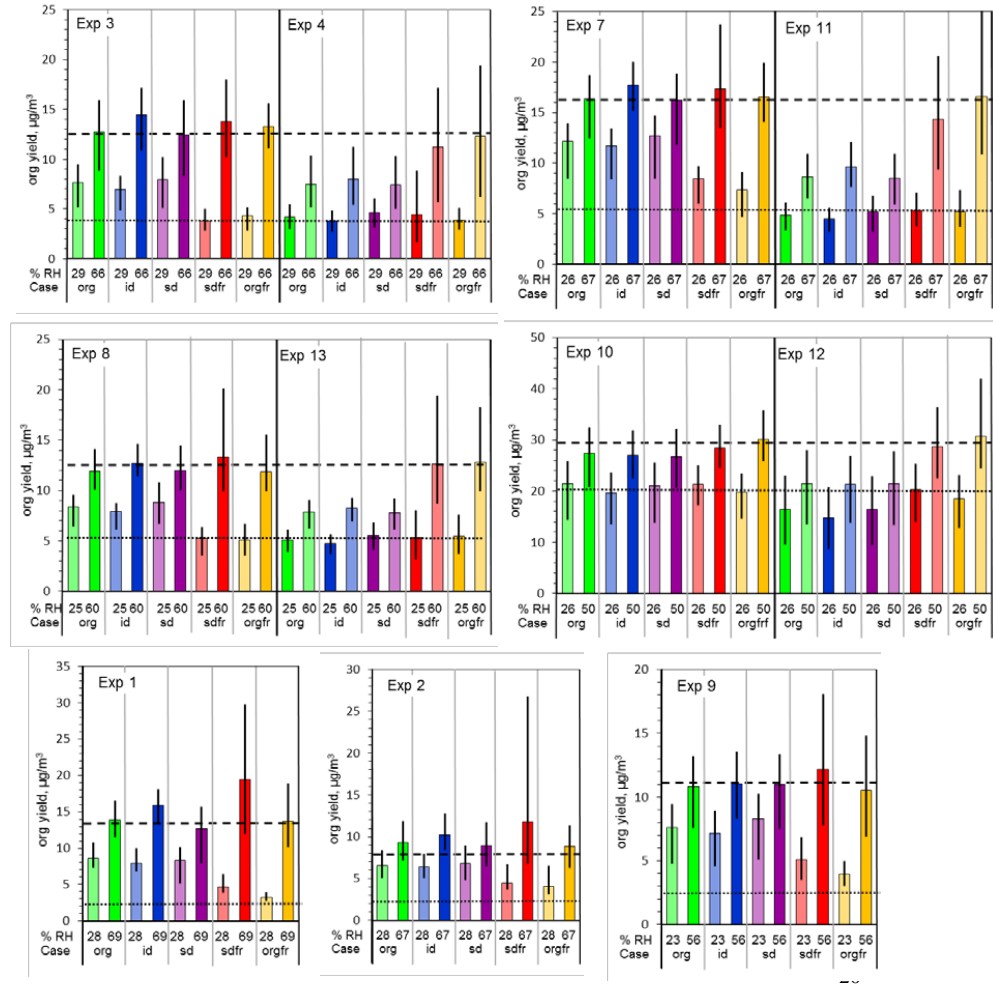

**Figure 7. Calculated average organic yields for experiments at the measured RH (columns in full colors) and at the RH of the corresponding experiments (light colors, see Table 2 for reference) based on ten randomly chosen volatility distributions for the cases org, id, sd, sdfr, and orgfr (identified by different colors). The vertical black lines on the columns indicate the range of values obtained for the calculations with the individual volatility distributions. The horizontal dotted line marks the measured value for the experiment performed at low RH, the dashed line the value for the experiment performed at high RH.**





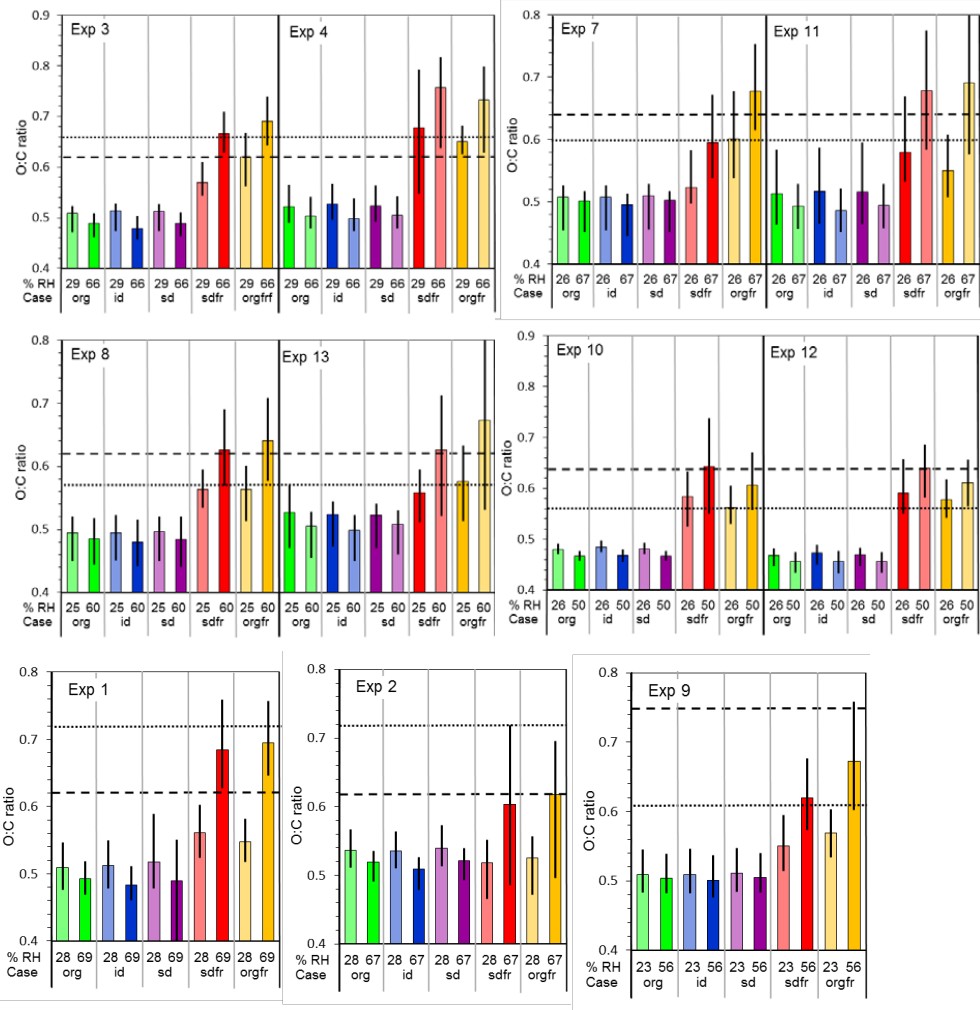

Figure 8. Calculated average O:C ratios for experiments at the measured RH (columns in full colors) and at the RH of the corresponding experiments (light colors, see Table 2 for reference) based on ten randomly chosen volatility distributions for the cases org, id, sd, sdfr, and orgfr (identified by different colors). The vertical black lines on the columns indicate the range of values obtained for the calculations with the individual volatility distributions. The horizontal dotted line marks the measured value for the experiment performed at low RH, the dashed line the value for the experiment performed at high RH.





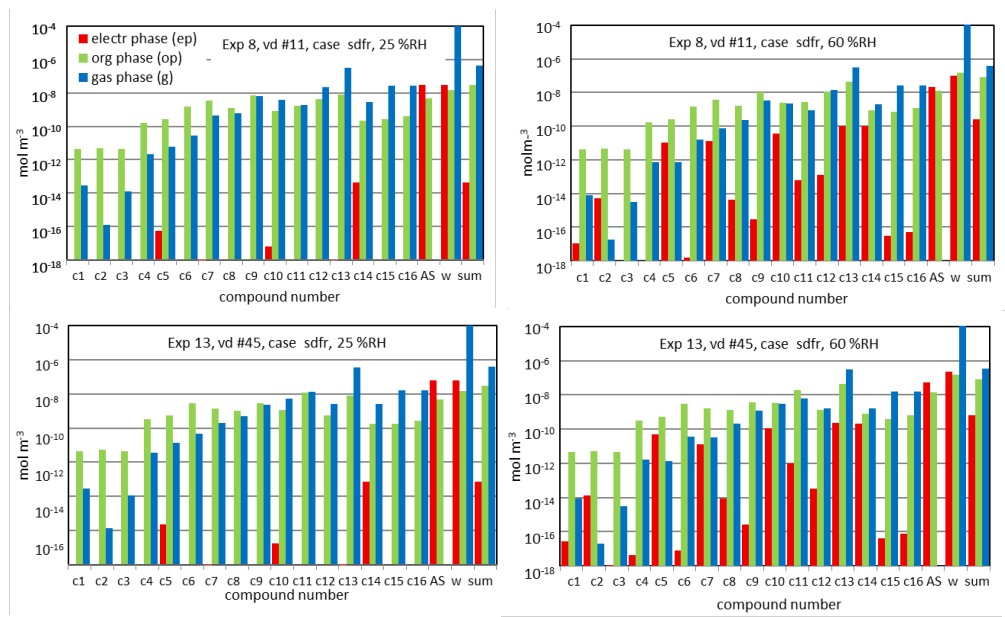

**Figure 9.** Equilibrium phase partitioning (mol m$^{-3}$) for case sdfr between gas phase, electrolyte phase, and organic phase at low RH (25 %) and at high RH (60 %) for experiment 8, volatility distribution #11 (upper panels) and experiment 13, volatility distribution #45 (lower panels). Compound numbers are c1: diaterpenylic acid acetate; c2: 3-MBTCA, c3: ValT4N9, c4: ValT4N10, c5: 3-hydroxyglutaric acid, c6: ValT4N3, c7: 3-oxoadipic acid; c8: pinic acid, c9: hopinonic acid, c10: glutaric acid: c11 norpinic acid, c12: 2-hydroxyterpenylic acid, c13: 5-COOH-3-OH-pentanal, c14: succinic acid, c15: 10-oxopinonic acid, c16: 4-oxopinonic acid, ammonium sulfate (AS), water (w), and the sum of compounds c1-c16.




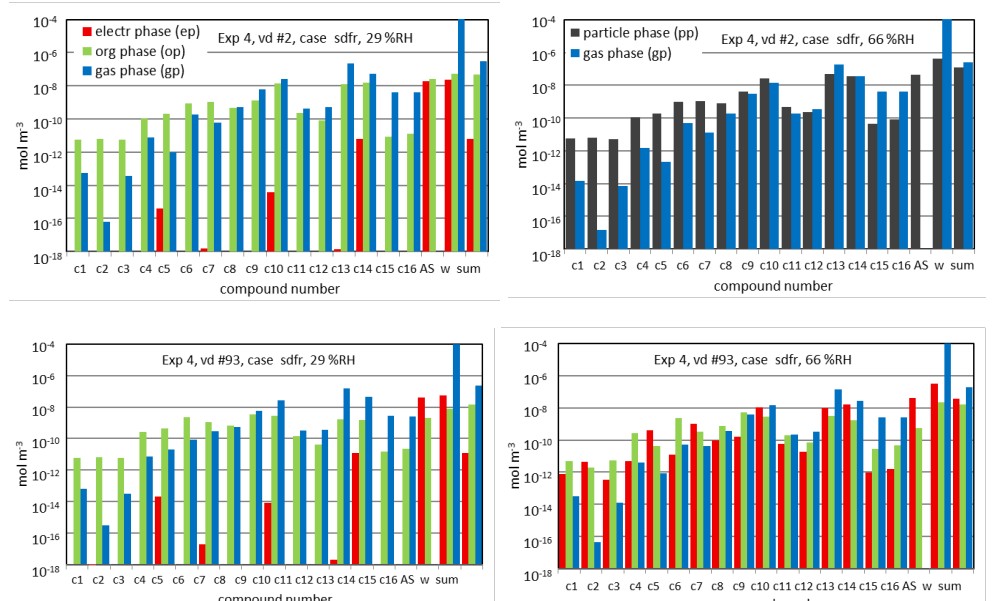

**Figure 10. Equilibrium phase partitioning (mol m⁻³) for case sdfr between gas phase, particle phase, electrolyte phase, and organic-rich phase at low (29 %) and high (66 %) RH for experiment 4, volatility distribution #2 (upper panels), and volatility distribution #93 (lower panels). Compound numbers are c1: diaterpenylic acid acetate; c2: 3-MBTCA, c3: ValT4N9, c4: ValT4N10, c5: 3-hydroxyglutaric acid, c6: ValT4N3, c7: 3-oxoadipic acid; c8: pinic acid, c9: hopinonic acid, c10: glutaric acid: c11 norpinic acid, c12: 2-hydroxyterpenylic acid, c13: 5-COOH-3-OH-pentanal, c14: succinic acid, c15: 10-oxopinonic acid, c16: 4-oxopinonic acid, ammonium sulfate (AS), water (w), and the sum of compounds c1-c16.**

50

55



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
