# Peer review of "$\alpha$ -pinene secondary organic aerosol yields increase at higher relative humidity and low NOx conditions"

_Atmospheric Chemistry and Physics, 2016_

## Referee Comment (RC1) · Anonymous Referee #1 · 11 Oct 2016

This work is laboratory and modeling study of SOA from alpha-pinene photooxidation under different RH conditions, and various NOx/VOC ratios. The authors investigated the effects of various seed compositions, notably hydrophobic vs. hydrophilic seed, which in my opinion is a clever way to deduce that the RH effect is significantly based on liquid water and particle phase mixtures. The authors primarily used AMS, PTRMS, and SMPS to measure the compounds/particles of interest and performed wall loss corrections for particles under one RH condition. Vapor wall loss was not considered. Appropriate blanks were performed. The work demonstrates a liquid-water-based enhancement of SOA yields that may be due to a combination of many chemical and physical factors. The authors also attempted to give insight into phase partition by using the AOIMFAC model and their own observational inputs. The paper is well-written and the method is thoroughly described. A more thorough discussion of the mechanisms involved, and additional clarity about the modeling, would be welcomed. I have some comments and suggestions before publication can be recommended.

General comments:

1. The authors stated that lack of corrections for vapor wall deposition do not "influence the comparison between the experiments." The statement is hard to understand when the RH dependence of vapor wall loss has been documented. For example, please see Loza et al EST (2010) and Nguyen et al PCCP (2016), where hydroperoxide, hydroxyepoxide, and organic acid wall losses were measured under different humidity conditions in chambers and differ substantially between different RH conditions. Nguyen et al PCCP (2016) even gave a parameterization for these compounds as a function of RH in a 24 cubic meter Teflon chamber (e.g., kwall_HMHP = $-1.4 \times 10-5 \times$ RH min$-1$, kwall_H2O2 = $-9.6 \times 10-6 \times$ RH min$-1$, and kwall_HCOOH = $-2.2 \times 10-6 \times$ RH min$-1$). As the authors can see, not only is vapor wall loss different for each RH condition, it is different for each chemical compound. The papers listed to support the authors' statement, namely Zhang et al PNAS (2014) and Nah et al ACP (2016), were only studied under dry (RH < 5%) conditions so are not applicable to the current case. I do not believe that retro-actively applying the vapor wall loss corrections is critical to this work, but request that the authors conservatively estimate the errors that ignoring such a correction in the alpha-pinene system (which is known to produce compounds readily lost to walls) would cause. This may actually increase the enhancement that the authors observed.

2. Adding to that subject, the authors are also suggested to monitor particulate wall losses at different RH conditions and for different composition in their future works in lieu of picking an average and going with it for all particles and all conditions. There is usually a noticeable difference in the rates of deposition depending on particle characteristics and wall wetness (related to RH). This is especially advisable since the authors

are using so many different seeds that may respond to wall wetness in different ways (I would guess the CF seeds would not show the same increase in sticking as the hydrophilic seeds when water layers on the walls increase, with similar effect to their SOA-derived observations). Again, the suggestion here is to approximate and report uncertainty that may be caused by ignoring these dependencies in the revised version of the paper.

3. I have a general criticism of the way "NOx" is used in this paper. It is too vague. When the authors say "44.4(0.8) ppbv of NOx" does that mean 42 ppb of NO2 and 2.4 ppb of NO or any of the other innumerable combinations...? Additionally, it's not really "NOx" that's important here, but rather nitric oxide (NO) because it changes the course of the reaction with the RO2 radical, while NO2 doesn't do very much unless the precursor is an aldehyde (which in this case it is not). Can the authors be more clear about how much NO there is, instead of "NOx"?

4. The authors mentioned hydrophilicity and solubility several times in the article, yet it's not clear how this is considered by the model, if at all? Also, despite the authors' statement that the few products considered in the model are adequate, more support is needed to understand how these few products can be fully representative of such a complex chemical system.

Some detailed comments

Pg 13, ln 12: It's not clear why the authors conclude that the NOx dependence is definitely due to low NOx conditions forming less volatile compounds? Perhaps the low-NOx conditions form more soluble compounds? Perhaps more well-mixed particles?

Pg 13, ln 18: How much of the reaction is actually ozonolysis? The authors should give an indication of ozone mixing ratio in these reactions, and calculate the prevalence of side reaction given the O3+a-pinene rates. If a significant fraction is ozonolysis, then RH will change the gas-phase product distribution as well.

[Figure]

Pg 14, ln 13: It would also be beneficial if the authors can talk about these products in terms of RO2 reactions. Also, why cite a 2011 modeling study when talking about hydroperoxide and carbonyl products from RO2 +HO2 and RO2+NO chemistry, when the mechanisms were deduced much earlier by Atkinson and many others and are now textbook knowledge?

Pg. 16, ln 17-20: The authors highlighted the importance of solubility in understanding the RH-dependent SOA yields, but the parameterizations only include volatility. The authors say later on that it's assumed that the hydrophilicity is proportional to the volatility, but that would mean treating a chemical process just like a physical process. Given that the model does not consider aqueous reactions, and how important these reactions have been shown to be for SOA (i.e., works of McNeill, Ervens, Carlton, and others), it's not clear to this reviewer that the augmented cases with fragmentation and lower volatility products (i.e., more volatility-driven solutions) give the right answers for the right reasons. How do the authors believe the modeling results would change if solubility and aqueous reactions were directly considered?

Table S4: Which chemical (i.e., NO, HO2) regime do these compounds belong to? Can the authors list the abundances that they derived, for each "NOx" regime? What were the hydrophilicity parameters that the authors assigned for these compounds?

---

## Referee Comment (RC2) · Anonymous Referee #2 · 17 Oct 2016

This manuscript presents experimental and modeling efforts in order to describe the formation of secondary organic aerosol (SOA) from photo-oxidation of alpha-pinene under various humidities and NOx concentrations. Also the role of seed aerosol composition is studied in relation to liquid water content of particles. The experiments clearly show that varying the above parameters cause large, and complex, changes in the SOA yields. The authors then attempt to draw further conclusions by phase partitioning calculations. While I find the model results to be less convincing, the model is described in detail, and thus readers can assess the validity of the different assumptions adequately. The paper fits the scope of ACP, and should be considered for publication following the below comments.

[Figure]

General comments:

While the language of the manuscript is very good, I found several descriptions and conclusions hard to follow. Especially the causality in certain sentences should be clarified. As an example, in the abstract it is stated "At low NOx conditions, equilibrium partitioning between the gas and liquid phases can explain most of the increase in SOA yields at high RH. This is indicated by the model results, when in addition to the $\alpha$-pinene photooxidation products described in the literature, more fragmented and oxidized organic compounds are added to the model mixtures". Is the point here that if adding oxidized fragments to the mixture (but not otherwise), the model can explain the increased yields at low NOx by equilibrium partitioning? The formulation of "indicated . . . when" is presumably the main reason for my confusion. Another example is page 15, lines 24-27: "Accordingly, additional insights into the prevalent mechanisms by which the compounds form and evolve can be gained. For example, highly oxygenated compounds cannot be very volatile without significant fragmentation, whereas oligomerization leads to a significant decrease in the compounds' vapor pressure without necessarily increasing their O:C ratios." It is unclear to me how the latter sentence is an insight gained from this work? And the content is in any case quite common knowledge, to some extent even used as an assumption in this work. There are several paragraphs with similar issues in the paper, and I recommend the authors (or preferably even someone external) read through the paper with the aim to check how claims of causality are presented.

Title: Currently, the title only reflects the experimental findings, while more focus is put on the model results in the text itself as well as the abstract. Also, the claim is left too general: is this true regardless of the oxidant (OH, ozone, nitrate radical) or [NO] (only NOx is mentioned). I suggest revising the title to better describe the content of the paper.

Specific comments:

P1, line 20: For terminology: SOA yields are not affected by particle wall loss. The measured SOA mass is, and if not accounting for that, one will get an *apparent* yield that is too low. On the other hand, vapor losses will affect the SOA yields in much more complicated ways. While I do not expect the authors to include vapor wall losses into the model at this stage, and it would be extremely hard to do correctly, the authors should acknowledge that there is a wealth of evidence from the last years that neglecting vapor wall loss will influence SOA yields, including e.g. Kokkola et al., 2014 (the first in a line of recent publications on the role of walls in Teflon chambers), Ehn et al., 2014 (detection of "ELVOC" that irreversibly are lost to walls) and Krechmer et al., 2016 (direct measurements of vapor wall losses in a Teflon chamber). The authors should at least note some of these papers and their findings in the manuscript, rather than only citing the papers that support their approach.

P1, line 20: "as a function of absorptive masses combining organics and the bound liquid water content." This is a confusing formulation. Rather say ". . .absorptive mass, defined as the sum of organics and the . . .".

P2, 18: Why limit this statement to semi-volatile species?

P5, 5-7: What does it mean when stating "similar NOx/VOC" when no a-pinene is added??

P7, 32-33: What were the ozone concentrations? I would like to see a (supplementary) figure with an example experiment showing at least a-pinene, ozone, butanol and OH concentrations together with the SOA mass.

Fig. 5: Why are figures 5 and 6 discussed before figures 3 and 4?

P12, 9: Please use another word than "corresponding" for these comparisons. It is misleading.

P13, 1-10: This is also consistent with more a-pinene producing more SOA and thereby condensation sink (CS), which in turn can more efficiently compete with the walls as a

sink of low-volatile vapors. See e.g. the papers cited above. The authors can easily do a sensitivity check for this effect. If there is a large change in particle vs wall loss rates, point 1 should be reconsidered. On the other hand, if the initial seed provides a relatively constant CS (compared to wall losses) for every experiment, then the claim in point 2 that seed concentrations in general are not important seems unjustified, since this was not probed at all in these experiments.

P13, 2: Please clearly distinguish between percentages and percentage points. I expect this should be the latter.

Fig. 4: There is a clear bimodal distribution for most cases, which is also noted in the manuscript. However, I find the explanations and discussion about it lacking. The authors state that particle number increased, but no new particle formation was observed. This needs some further discussion. Where do the particles come from then? Additionally, the bimodality is used as proof for LLPS, which the model also predicts, but I do not find a clear description of why the organics form this bimodal distribution.

Fig. 7&8: Make the contrast between the currently light and full colors more visible. At least on my screen some pairs were hard to distinguish.

P15, 36-37 and related O:C discussion: A variation of 0.03 from 0.45 to 0.48 is considered "almost constant" while an increase of 0.08 is considered significant? And only several pages later in section 4.6 is it noted that the uncertainty in O:C is 20-30%. It is also noted that the O:C values are likely biased low since the latest parametrization for O:C calculations are not used. These things should be mentioned earlier, so a reader can properly assess the meaningfulness of the comparisons done in sections 4.2-4.3. Considering all the above, the tuning of the model to match these values does not in my mind give much more insight into the formation mechanisms of SOA in this system.

P18, 3-5: Is this shown somewhere, or just stated? There were also other places where the formulations are such that I expect there to be a figure showing the result. The authors should consider adding "not shown" in places where the information is not

visible in any plot or table.

P18, 22-26. There are too many numbers listed in the text, and this is especially true here. Please consider rewriting this.

P18, 19-22 and Fig. 9: It is hard to follow discussion about increase of factors 2-5 from a plot ranging 14 orders of magnitude. Could Fig. 9 be moved to the SI, and some more specific plot included in the main text?

P18, 39-40: I do not understand at all what this sentence is supposed to say.

P19, 7: "sufficient" for what?

P20, 22: Expected based on what?

P21, 28-29: This is too strong a statement in my opinion. Rather say that only with inclusion of the fragments could your model describe both SOA mass and O:C.

P21, 40-P22, 2: Such a statement should be included much earlier in the discussions on bimodality, and not saved to the last lines of the manuscript.

P22, 2: Again, what is this expectation based on? Work in this paper or other work?

References

Ehn, M., et al. (2014). Nature, 506(7489), 476-479.

Kokkola, H., et al. (2014). Atmospheric Chemistry and Physics, 14(3), 1689-1700.

Krechmer, J. E., et al. (2016). Environmental Science & Technology, 50(11), 5757-5765.

---

## Author Comment (AC1) · 6 Jan 2017

Author comments to the manuscript: "α-Pinene secondary organic aerosol yields increase at higher relative humidity and low $NO_x$ conditions" by Lisa Stirnweis et al.

L. Stirnweis et al.

The authors would like to thank the two anonymous reviewers for their helpful and thoughtful comments and suggestions. Based on their comments we have modified the text, especially the parts related to the model description, $NO_x$ regime and vapour wall losses, where additional explanations seemed to be necessary. Each specific point of the comments is addressed below. Reviewer comments are in italic typeset, our responses are in regular typeset and changes in the manuscript are highlighted in blue. All references are listed at the end of the document.

**Anonymous Referee #1**

*This work is laboratory and modeling study of SOA from alpha-pinene photooxidation under different RH conditions, and various NOx/VOC ratios. The authors investigated the effects of various seed compositions, notably hydrophobic vs. hydrophilic seed, which in my opinion is a clever way to deduce that the RH effect is significantly based on liquid water and particle phase mixtures. The authors primarily used AMS, PTRMS, and SMPS to measure the compounds/particles of interest and performed wall loss corrections for particles under one RH condition. Vapor wall loss was not considered. Appropriate blanks were performed. The work demonstrates a liquid-water-based enhancement of SOA yields that may be due to a combination of many chemical and physical factors. The authors also attempted to give insight into phase partition by using the AOIMFAC model and their own observational inputs. The paper is well-written and the method is thoroughly described. A more thorough discussion of the mechanisms involved, and additional clarity about the modeling, would be welcomed. I have some comments and suggestions before publication can be recommended.*

*General comments:*

*1. The authors stated that lack of corrections for vapor wall deposition do not "influence the comparison between the experiments." The statement is hard to understand when the RH dependence of vapor wall loss has been documented. For example, please see Loza et al EST (2010) and Nguyen et al PCCP (2016), where hydroperoxide, hydroxyepoxide, and organic acid wall losses were measured under different humidity conditions in chambers and differ substantially between different RH conditions. Nguyen et al PCCP (2016) even gave a*

*parameterization for these compounds as a function of RH in a 24 cubic meter Teflon chamber (e.g., kwall_HMHP = −1.4 × 10−5 × RH min−1, kwall_H2O2 = −9.6 × 10−6 × RH min−1, and kwall_HCOOH = −2.2 × 10−6 × RH min−1). As the authors can see, not only is vapor wall loss different for each RH condition, it is different for each chemical compound. The papers listed to support the authors' statement, namely Zhang et al PNAS (2014) and Nah et al ACP (2016), were only studied under dry (RH < 5%) conditions so are not applicable to the current case. I do not believe that retro-actively applying the vapor wall loss corrections is critical to this work, but request that the authors conservatively estimate the errors that ignoring such a correction in the alpha-pinene system (which is known to produce compounds readily lost to walls) would cause. This may actually increase the enhancement that the authors observed.*

**Response:** We have already acknowledged in the original version of the manuscript that vapour wall losses may influence chamber results and complicate data interpretation. This is especially the case if losses depend on the RH, as these losses would lead to biases that were not yet, but will be discussed in the manuscript. Nevertheless, we note that losses of the small organic molecules reported in Loza et al. (2010) and Nguyen et al. (2016), e.g. glyoxal, epoxides and peroxides are very likely due to their reactive uptake, the rate of which changes with RH. These processes occur at time scales of hours, much longer than those related to the absorption of SOA forming semi-volatile compounds onto the clean Teflon walls. The dependence of vapour absorption on RH (due to a change in wall accommodation coefficients or in the activity of the wall absorbed compounds) has not been reported to the best of our knowledge and indeed merits further investigations that are beyond the scope of the current study. Nevertheless, we believe that vapour absorption onto the Teflon walls is unlikely to be affected by RH, due to the hydrophobic nature of Teflon and its minor interactions with water at relatively low RH (<75%).

Based on the reviewer suggestions, we have discussed these effects more thoroughly in the revised version of the manuscript:

Eq. (8) does not take into consideration the loss of SOA-forming vapours onto the clean Teflon walls, which may suppress SOA yields from laboratory chambers under certain conditions. These processes may be related to the vapours reactive uptake onto walls (Loza et al., 2010; Nguyen et al., 2016) or to their absorptive uptake (Zhang et al., 2014; Nah et al., 2016).

The reactive uptake of organic vapours onto chamber walls is only significant at high RH. Loss rates for important reactive gases, including glyoxal, epoxides and peroxides have been documented at different RH (Loza et al., 2010; Nguyen et al., 2016)While these processes may also influence reactive SOA-forming compounds under our conditions, they occur at time-scales of hours (Nguyen et al., 2016), much longer compared to the time scales of the absorptive uptake, e.g. based on recent direct measurements of vapour losses onto Teflon walls (~10 min,(Krechmer et al., 2016)).

The absorption of organic compounds onto the chamber walls obeys Henry's law and depends on the compounds' accommodation coefficients and their activity at the wall/gas interface (see for example, (Zhang et al., 2015)). The dependence of compounds' absorption on RH (due to a change in accommodation coefficients or in the activity of the wall absorbed compounds) has not yet been reported to the best of our knowledge and indeed merits further investigations that are beyond the scope of the current study. Nevertheless, we believe that vapour absorption onto the walls is unlikely to be significantly affected by RH, due to the hydrophobic nature of Teflon and its minor interaction with water under subsaturation conditions (RH<80%).

In our case, we have maintained chamber conditions during our experiments such that vapour wall losses and their inter-experimental differences can be minimized as much as possible. This is done by (1) maintaining a relatively constant wall-to-seed surface ratio for all experiments to avoid systematic biases between experiments and (2) increasing SOA production rates, which rapidly provide a significant particle condensational sink. We also note that vapour wall losses were found to be minor for the α-pinene SOA system where SOA formation is dominated by quasi-equilibrium growth (Zhang et al., 2014; Nah et al., 2016).

*2. Adding to that subject, the authors are also suggested to monitor particulate wall losses at different RH conditions and for different composition in their future works in lieu of picking an average and going with it for all particles and all conditions. There is usually a noticeable difference in the rates of deposition depending on particle characteristics and wall wetness (related to RH). This is especially advisable since the authors are using so many different seeds that may respond to wall wetness in different ways (I would guess the CF seeds would not show the same increase in sticking as the hydrophilic seeds when water layers on the walls increase, with similar effect to their SOA-derived observations). Again, the suggestion here is to approximate and report uncertainty that may be caused by ignoring these dependencies in the revised version of the paper.*

**Response:** Unlike vapours, particles are lost to the walls irreversibly and with an accommodation coefficient of unity. Therefore, particle wall losses, driven by diffusion and gravitation, are less dependent on the wall wetness (only through a change in the Teflon electrostatic properties with RH), and are strongly a function of particle diameters. As the organic particles and inorganic seeds are externally mixed and have different diameters, we have chosen to derive the loss rates of the organic phase by fitting its decay with time (instead of using the seed decay), when OA production is expected to be negligible. We have derived a loss rate for each experiment, which was used for wall loss correction. An average rate was only used in the case of insufficient statistics to perform accurate fitting. Therefore, we believe that our wall loss procedure takes into account the experiment-to-experiment variability.

*3. I have a general criticism of the way "NOx" is used in this paper. It is too vague. When the authors say "44.4(0.8) ppbv of NOx" does that mean 42 ppb of NO2 and 2.4 ppb of NO or any of the other innumerable combinations. . .? Additionally, it's not really "NOx" that's important here, but rather nitric oxide (NO) because it changes the course of the reaction with the RO2 radical, while NO2 doesn't do very much unless the precursor is an aldehyde (which in this case it is not). Can the authors be more clear about how much NO there is, instead of "NOx"?*

**Response:** Based on our previous calculations (see Platt et al. (2014)), $RO_2$ radicals would predominantly react with NO, when the concentration of the latter is higher than only 1 ppb. These conditions can be considered as high $NO_x$.

As mentioned in the manuscript, during high $NO_x$ experiments, before lights were turned on, we have injected equal amounts of NO and $NO_2$, resulting in initial $NO_x$ concentrations between 20-75 ppb. During these experiments and throughout the period when the majority of α-pinene was consumed, the NO concentration remained higher than 5ppb, indicating that α-pinene oxidation proceeded under high $NO_x$.

During low $NO_x$ experiments, the average $NO_x$ concentration was around 1-2 ppb, predominated by $NO_2$ while the NO concentration was below detection limits (<0.1ppb). Under these conditions $RO_2$-$RO_2$ reactions may prevail.

Based on the reviewer comment, we have updated the text on page 5, as follows:

Within the results section, the two terms "low $NO_x$" and "high $NO_x$" refer to the following conditions:

- Low $NO_x$ = $NO_x$/α-pinene < 0.1, with continuous HONO injection, indicated by an asterisk in Table 1 and figures.

- High $NO_x$ = $NO_x$/α-pinene > 1: Initial injection of NO + $NO_2$ with continuous HONO injection.

In Table 1, we report for high $NO_x$ conditions the initial $NO_x$ concentration, (which decays with time) and for low $NO_x$ conditions the mean $NO_x$ concentration. We note that the NO levels are the main driver for determining whether $RO_2$-NO or $RO_2$-$RO_2$ reactions would prevail. Based on our calculations (not shown here, see (Platt et al., 2014)), $RO_2$ radicals would predominantly react with NO, when the concentration of the latter is higher than only 1 ppb. These conditions can be considered as high $NO_x$.

During high $NO_x$ experiments and throughout the period when the majority of α-pinene was consumed, the NO concentration remained higher than 5ppb, indicating that α-pinene oxidation proceeded under high $NO_x$. During low $NO_x$ experiments, the average $NO_x$ concentration was around 1-2 ppb, predominated by $NO_2$, while the NO concentrations were below detection limits (<0.1ppb). Under these conditions $RO_2$-$RO_2$ reactions may prevail.

*4. The authors mentioned hydrophilicity and solubility several times in the article, yet it's not clear how this is considered by the model, if at all? Also, despite the authors' statement that the few products considered in the model are adequate, more support is needed to understand how these few products can be fully representative of such a complex chemical system.*

**Response:** The aim of the modelling part is to investigate whether thermodynamics alone can account for the higher yields observed at higher RH. The compounds used for the simulations were chosen from previously identified photo-oxidation products of α-pinene. The partitioning of these compounds between the gas and the condensed phases depends on their effective saturation concentrations, which is defined in Eq. 11. This equation takes solubility and hydrophilicity into account because it does not depend only on the vapour pressures of the pure compounds but also on their activity coefficients in solution (i.e. affinity towards other compounds in the organic phase, affinity towards water and affinity towards the inorganic species). We use AIOMFAC to calculate activity coefficients and perform a phase equilibration calculation between the gas and condensed phases. With this approach, we fully account for solubility (in the organic phase) and hydrophilicity (affinity towards particulate water).

As we have mentioned in the manuscript, the choice of the model compounds is constrained by the experimentally determined volatility distributions and O:C ratios. This significantly

constrains the nature of the compounds in a given volatility bin. For example, compounds with saturation concentrations of 100-1000 µg m$^{-3}$ would need to have a short carbon backbone chain (carbon number ~5), if their O:C ratios should match the observed O:C ratios of ~0.6. The activity coefficients calculated by the model are more sensitive to the chosen compounds O:C ratios than to the compounds' structure and functional groups.

From the comments/questions of both reviewers, further details and clarifications seem to be necessary for a better explanation of the procedure adopted for the phase partitioning calculations. Therefore, in the corrected version of the manuscript, we have significantly modified section 2.6, relative to phase partitioning calculations.

Page 10, lines 12-29 were replaced by the following:

**2.6 Thermodynamic modelling**

[revised manuscript text omitted]

**Some detailed comments**

*Pg 13, ln 12: It's not clear why the authors conclude that the NOx dependence is definitely due to low NOx conditions forming less volatile compounds? Perhaps the low- NOx conditions form more soluble compounds? Perhaps more well-mixed particles?*

**Response:** Conclusions on the NO$_x$ dependence are based on the multilinear analysis results in section 3.1. We have observed that independent of the prevailing RH, yields obtained under low NO$_x$ are significantly higher than those obtained under high NO$_x$ conditions. This implies that independent of the compounds affinity towards water, compounds formed under low NO$_x$ are less volatile than those obtained under high NO$_x$ and tend to remain in the particle phase.

*Pg 13, ln 18: How much of the reaction is actually ozonolysis? The authors should give an indication of ozone mixing ratio in these reactions, and calculate the prevalence of side reaction given the O3+a-pinene rates. If a significant fraction is ozonolysis, then RH will change the gas-phase product distribution as well.*

**Response:** In the original version of the manuscript, we have already assessed and discussed the fraction of α-pinene that reacted with OH and ozone (see section 2.4 and table 1), based on the mixing ratios of these two oxidants. We have estimated that a great part of α-pinene reacted with OH (on average $0.78 \pm 0.07$) and noted that the further processing of the first generation products – which don't contain C=C bonds – would almost exclusively proceed through OH oxidation. Accordingly, we concluded that SOA compounds detected under our conditions are mainly from OH chemistry.

We note that the fraction of α-pinene that reacted with OH under low RH (0.79 ± 0.07) and high RH (0.77 ± 0.07) are not statistically different (t-test, $p$=0.71), within our experimental variability. Therefore, we do not expect that differences observed between experiments at low and high RH are due to a change in the prevalent oxidant. We recognize that water vapor may change the oxidation product distributions, via its reaction with the stabilized Criegee intermediates, produced upon the ozonolysis of α-pinene. However, we note that not only the fraction of α-pinene that reacts with $O_3$ is not substantial (in comparison with that reacted with OH), but also a major fraction of α-pinene Criegees undergoes unimolecular decomposition to form OH and does not react with water (Atkinson and Arey, 2003). Therefore, it is unlikely that a change in RH would sizably modify the distribution of the products formed via gas-phase chemistry.

By contrast, the fraction of α-pinene that reacted with OH is found to be sensitive to the $NO_x$ concentration (as already mentioned in the manuscript). The production of $O_3$ was faster under high $NO_x$ compared to low $NO_x$, due to an efficient VOC-$NO_x$ catalytic cycle and a higher fraction of OH was scavenged by $NO_2$. Consequently, the fraction of α-pinene that reacted with OH under high $NO_x$ (0.75 ± 0.06) is lower than that under low $NO_X$ (0.83 ± 0.04). These small, but statistically significant (0.08; t-test, $p$=0.004) differences in the contribution of ozone/OH to α-pinene oxidation are expected to explain a part of the differences in SOA yields and chemical composition at low and high $NO_x$, with higher fraction of ozonolysis products under high $NO_x$ conditions.

While some of the above information was discussed in the old version of the manuscript, we have modified the new text by adding a new section (3.4) related to the above discussion and modified the old section 2.4, accordingly:

**3.4 Prevalent oxidation reagent and its influence on SOA yields and chemical composition**

Based on the mixing ratios of OH and ozone, we estimate that a great part of α-pinene has reacted with OH (on average 0.78 ± 0.07). Moreover, it is worthwhile to mention that the further processing of the first generation products – which do not contain C=C bonds – would almost exclusively proceed through OH oxidation. Accordingly, we conclude that SOA compounds detected are mainly from OH chemistry independent of the $NO_x$ level and relative humidity.

We note that the fraction of α-pinene that reacted with OH under low RH (0.79 ± 0.07) and high RH (0.77 ± 0.07) are not statistically different (t-test, $p$=0.71), within our experimental variability. Therefore, we do not expect that differences in SOA yields observed between

experiments at low and high RH to be due to a change in the prevalent oxidant. We recognize that water vapor may change the oxidation product distributions, via its reaction with the stabilized Criegee intermediates, produced upon the ozonolysis of α-pinene. However, we note that not only the fraction of α-pinene that reacts with $O_3$ is not substantial (in comparison with that reacted with OH), but also a major fraction of α-pinene Criegee intermediates undergoes unimolecular decomposition to form OH and does not react with water (Atkinson and Arey, 2003). Therefore, it is unlikely that a change in RH would sizably modify the distribution of the products formed via gas-phase chemistry.

By contrast, the fraction of α-pinene that reacted with OH is found to be sensitive to the $NO_x$ concentration. The production of $O_3$ was faster under high $NO_x$ (average $[O_3]$ = 35 ppb) compared to low $NO_x$ (average $[O_3]$ = 22 ppb), due to an efficient VOC-$NO_x$ catalytic cycle and a higher fraction of OH was scavenged by $NO_2$. Consequently, the fraction of α-pinene that reacted with OH under high $NO_x$ (0.75 ± 0.06) is lower than that under low $NO_x$ (0.83 ± 0.04). These small, but statistically significant differences (0.08; t-test, $p$=0.004) in the contribution of ozone/OH to α-pinene oxidation are expected to explain a small part of the differences in SOA yields and chemical composition observed at low and high $NO_x$, with higher fraction of ozonolysis products under high $NO_x$ conditions. Despite this, we expect the influence of $NO_x$ on the fate of $RO_2$ to be the main driver behind the observed differences in SOA yields and chemical composition between low and high $NO_x$ conditions (because ozonolysis products are only a minor fraction and differences between the two conditions are rather small).

*Pg 14, ln 13: It would also be beneficial if the authors can talk about these products in terms of RO2 reactions. Also, why cite a 2011 modeling study when talking about hydroperoxide and carbonyl products from RO2 +HO2 and RO2+NO chemistry, when the mechanisms were deduced much earlier by Atkinson and many others and are now textbook knowledge?*

**Response:** As we have mentioned above, model compounds are only surrogates. While we recognize that oxidation conditions, e.g. $NO_x$ concentrations, may significantly alter the product distribution, we did not select different sets of these surrogates for the different conditions. This is because:

(1) such separation would implicitly suggest that the chemical composition of the few compounds reported at different conditions can be extrapolated to the bulk OA under our conditions;

(2) such separation would significantly limit the number of surrogates at each condition, increasing the sensitivity of the model to the compounds' selection; and

(3) the model is less sensitive to compounds' chemical structure than to their elemental composition (see below and Li et al. (2016)), e.g. number of oxygen and carbon, which has been taken into account by including fragmentation products.

Instead, the same compounds reported at low and high $NO_x$ are used together with fragmentation products as model inputs for all experiments. The contribution of these products is optimized in the model, so that the modelled SOA yields and O:C ratios at a given RH matches the observations. Therefore, while explicit molecular structures are needed as model inputs, the contribution of these molecules in the model is not expected to reflect their true contribution to SOA at different RH and oxidation conditions, especially based on the way the model is setup. Accordingly, we think that discussing differences in the model compounds' contribution at different conditions would not be suitable.

Based on the reviewer comment, we have replaced the citation of a modelling study, Valorso et al. (2011), by the earlier studies reviewed in (Atkinson, 2000), when discussing the influence of $NO_x$ on SOA bulk composition (page 14 line 13).

*Pg. 16, ln 17-20: The authors highlighted the importance of solubility in understanding the RH-dependent SOA yields, but the parameterizations only include volatility. The authors say later on that it's assumed that the hydrophilicity is proportional to the volatility, but that would mean treating a chemical process just like a physical process. Given that the model does not consider aqueous reactions, and how important these reactions have been shown to before SOA (i.e., works of McNeill, Ervens, Carlton, and others), it's not clear to this reviewer that the augmented cases with fragmentation and lower volatility products (i.e., more volatility-driven solutions) give the right answers for the right reasons. How do the authors believe the modeling results would change if solubility and aqueous reactions were directly considered?*

**Response:** The model takes hydrophilicity and solubility into account through the activity coefficients of the partitioning compounds. Condensed phase reactions are not considered, nor the possibility that the reactivity depends on relative humidity. We use the thermodynamic simulations to find out whether partitioning alone can account for the increased yield at high RH. Our results show that at low $NO_x$ conditions, equilibrium partitioning between the gas and liquid phases can explain most of the increase in SOA yields at high RH. In contrast, at

high $NO_x$, equilibrium partitioning alone could not explain the strong increase in the yields with increased RH. Therefore we conclude that under these conditions additional processes including the reactive uptake of semi-volatile species in the particulate aqueous phase would need to occur to explain the enhanced yield at higher RH.

*Table S4: Which chemical (i.e., NO, HO2) regime do these compounds belong to? Can the authors list the abundances that they derived, for each "NOx" regime? What were the hydrophilicity parameters that the authors assigned for these compounds?*

**Response:** We cannot assign hydrophilicity parameters to model compounds, because compounds' activity coefficients, which determine their solubility and hydrophilicity, depend on the composition of the condensed phases and therefore are obtained iteratively during the phase equilibration calculation. As mentioned above, while explicit molecular structures are needed as model inputs, the way the model is setup does not allow us to discuss the influence of the chemical regime on the compounds distributions.

**Anonymous Referee #2**

*This manuscript presents experimental and modelling efforts in order to describe the formation of secondary organic aerosol (SOA) from photo-oxidation of alpha-pinene under various humidities and NOx concentrations. Also the role of seed aerosol com- position is studied in relation to liquid water content of particles. The experiments clearly show that varying the above parameters cause large, and complex, changes in the SOA yields. The authors then attempt to draw further conclusions by phase partitioning calculations. While I find the model results to be less convincing, the model is described in detail, and thus readers can assess the validity of the different assumptions adequately. The paper fits the scope of ACP, and should be considered for publication following the below comments.*

**General comments:**

*While the language of the manuscript is very good, I found several descriptions and conclusions hard to follow. Especially the causality in certain sentences should be clarified. As an example, in the abstract it is stated "At low NOx conditions, equilibrium partitioning between the gas and liquid phases can explain most of the increase in SOA yields at high RH. This is indicated by the model results, when in addition to the α-pinene photooxidation products described in the literature, more fragmented and oxidized organic compounds are added to the model mixtures". Is the point here that if adding oxidized fragments to the mixture (but not otherwise), the model can explain the increased yields at low NOx by equilibrium partitioning? The formulation of "indicated ... when" is presumably the main reason for my confusion. Another example is page 15, lines 24-27: "Accordingly, additional insights into the prevalent mechanisms by which the compounds form and evolve can be gained. For example, highly oxygenated compounds cannot be very volatile without significant fragmentation, whereas oligomerization leads to a significant decrease in the compounds' vapor pressure with- out necessarily increasing their O:C ratios." It is unclear to me how the latter sentence is an insight gained from this work? And the content is in any case quite common knowledge, to some extent even used as an assumption in this work. There are several paragraphs with similar issues in the paper, and I recommend the authors (or preferably even someone external) read through the paper with the aim to check how claims of causality are presented.*

*Title: Currently, the title only reflects the experimental findings, while more focus is put on the model results in the text itself as well as the abstract. Also, the claim is left too general: is this*

*true regardless of the oxidant (OH, ozone, nitrate radical) or [NO] (only NOx is mentioned). I suggest revising the title to better describe the content of the paper.*

**Response:** Based on both reviewers' comments, we have significantly modified some parts of the text, especially those related to the model description section. In addition, we have modified the abstract, based on the above comment, as follows:

At low $NO_x$ conditions, equilibrium partitioning between the gas and liquid phases can explain most of the increase in SOA yields observed at high RH, when in addition to the α-pinene photo-oxidation products described in the literature, fragmentation products are added to the model mixtures. This increase is driven by both the increase in the absorptive mass and the solution non-ideality described by compounds' activity coefficients. In contrast, at high $NO_x$, equilibrium partitioning alone could not explain the strong increase in the yields with RH. This suggests that other processes, e.g. reactive uptake of semi-volatile species into the liquid phase, may occur and be enhanced at higher RH, especially for compounds formed under high $NO_x$ conditions, e.g. carbonyls.

Based on the comment above, we also modified the sentence on page 15, lines 24-27:

Nevertheless, fitting both organic yields and O:C ratios significantly aid constraining the type of compounds that participate in partitioning (i.e. from a compound O:C ratio and vapor pressure its carbon number can be inferred). For example, highly oxygenated compounds cannot be very volatile without significant fragmentation, whereas oligomerization leads to a significant decrease in the compounds' vapor pressure without necessarily increasing their O:C ratios.

We have also modified the title as follows:

Assessing the Influence of $NO_x$ Concentrations and Relative Humidity on Secondary Organic Aerosol Yields from α-Pinene photo-oxidation through Smog Chamber Experiments and Modelling Calculations

In the text we have also discussed more thoroughly the influence of the oxidants and $NO_x$ on our conclusions (see above).

*Specific comments:*

*P1, line 20: For terminology: SOA yields are not affected by particle wall loss. The measured SOA mass is, and if not accounting for that, one will get an \*apparent\* yield that is too low.*

*On the other hand, vapor losses will affect the SOA yields in much more complicated ways. While I do not expect the authors to include vapor wall losses into the model at this stage, and it would be extremely hard to do correctly, the authors should acknowledge that there is a wealth of evidence from the last years that neglecting vapor wall loss will influence SOA yields, including e.g. Kokkola et al., 2014 (the first in a line of recent publications on the role of walls in Teflon chambers), Ehn et al., 2014 (detection of "ELVOC" that irreversibly are lost to walls) and Krechmer et al., 2016 (direct measurements of vapor wall losses in a Teflon chamber). The authors should at least note some of these papers and their findings in the manuscript, rather than only citing the papers that support their approach.*

**Response:** Based on the reviewer comment the sentence on P1, line 20, has been modified in the corrected version of the manuscript as follows:

We used a Monte-Carlo approach to parameterize smog chamber SOA yields as a function of the condensed phase absorptive mass, which includes the sum of OA and the corresponding bound liquid water content.

Also, we have already cited four papers related to wall-losses, but we will add the references suggested by both reviewers. We have significantly changed the section related to the vapour wall losses, where these additional citations were added. The new text reads as follows:

Eq. (8) does not take into consideration the loss of SOA-forming vapours onto the clean Teflon walls, which may suppress SOA yields from laboratory chambers under certain conditions. These processes may be related to the vapours reactive uptake onto walls (Loza et al., 2010; Nguyen et al., 2016) or to their absorptive uptake (Zhang et al., 2014; Nah et al., 2016).

The reactive uptake of organic vapours onto chamber walls is only significant at high RH. Loss rates for important reactive gases, including glyoxal, epoxides and peroxides have been documented at different RH (Loza et al., 2010; Nguyen et al., 2016)While these processes may also influence reactive SOA-forming compounds under our conditions, they occur at time-scales of hours (Nguyen et al., 2016), much longer compared to the time scales of the absorptive uptake, e.g. based on recent direct measurements of vapour losses onto Teflon walls (~10 min,(Krechmer et al., 2016)).

The absorption of organic compounds onto the chamber walls obeys Henry's law and depends on the compounds' accommodation coefficients and their activity at the wall/gas interface (see for example, (Zhang et al., 2015)). The dependence of compounds' absorption on RH (due to a change in accommodation coefficients or in the activity of the wall absorbed

compounds) has not yet been reported to the best of our knowledge and indeed merits further investigations that are beyond the scope of the current study. Nevertheless, we believe that vapour absorption onto the walls is unlikely to be significantly affected by RH, due to the hydrophobic nature of Teflon and its minor interaction with water under subsaturation conditions (RH<80%).

In our case, we have maintained chamber conditions during our experiments such that vapour wall losses and their inter-experimental differences can be minimized as much as possible. This is done by (1) maintaining a relatively constant wall-to-seed surface ratio for all experiments to avoid systematic biases between experiments and (2) increasing SOA production rates, which rapidly provide a significant particle condensational sink. We also note that vapour wall losses were found to be minor for the α-pinene SOA system where SOA formation is dominated by quasi-equilibrium growth (Zhang et al., 2014; Nah et al., 2016).

*P1, line 20: "as a function of absorptive masses combining organics and the bound liquid water content." This is a confusing formulation. Rather say "...absortive mass, defined as the sum of organics and the ...".*

**Response:** This has been modified in the revised version of the manuscript as follows:

We used a Monte-Carlo approach to parameterize smog chamber SOA yields as a function of the condensed phase absorptive mass, which includes the sum of OA and the corresponding bound liquid water content.

*P2, 18: Why limit this statement to semi-volatile species?*

**Response:** In the revised version of the manuscript we remove "semi-volatile"

*P5, 5-7: What does it mean when stating "similar NOx/VOC" when no a-pinene is added??*

**Response:** $NO_x$/VOC was replaced by $NO_x$.

*P7, 32-33: What were the ozone concentrations? I would like to see a (supplementary) figure with an example experiment showing at least a-pinene, ozone, butanol and OH concentrations together with the SOA mass. Fig. 5: Why are figures 5 and 6 discussed before figures 3 and 4?*

**Response:** $O_3$ average concentrations were 22 and 35ppb at low $NO_x$ and high $NO_x$, respectively. In the revised version of the manuscript we have modified section 2.4 (P7, lines 32-33), and added an entire new section, section 3.4, where we discuss the relative importance of O3 and OH for the oxidation of α-pinene at different conditions:

**3.4 Prevalent oxidation reagent and its influence on SOA yields and chemical composition**

Based on the mixing ratios of OH and ozone, we estimate that a great part of α-pinene has reacted with OH (on average $0.78 \pm 0.07$). Moreover, it is worthwhile to mention that the further processing of the first generation products – which do not contain C=C bonds – would almost exclusively proceed through OH oxidation. Accordingly, we conclude that SOA compounds detected are mainly from OH chemistry independent of the $NO_x$ level and relative humidity.

We note that the fraction of α-pinene that reacted with OH under low RH ($0.79 \pm 0.07$) and high RH ($0.77 \pm 0.07$) are not statistically different (t-test, $p=0.71$), within our experimental variability. Therefore, we do not expect that differences in SOA yields observed between experiments at low and high RH to be due to a change in the prevalent oxidant. We recognize that water vapor may change the oxidation product distributions, via its reaction with the stabilized Criegee intermediates, produced upon the ozonolysis of α-pinene. However, we note that not only the fraction of α-pinene that reacts with $O_3$ is not substantial (in comparison with that reacted with OH), but also a major fraction of α-pinene Criegee intermediates undergoes unimolecular decomposition to form OH and does not react with water (Atkinson and Arey, 2003). Therefore, it is unlikely that a change in RH would sizably modify the distribution of the products formed via gas-phase chemistry.

By contrast, the fraction of α-pinene that reacted with OH is found to be sensitive to the $NO_x$ concentration. The production of $O_3$ was faster under high $NO_x$ (average $[O_3] = 35$ ppb) compared to low $NO_x$ (average $[O_3] = 22$ ppb), due to an efficient VOC-$NO_x$ catalytic cycle and a higher fraction of OH was scavenged by $NO_2$. Consequently, the fraction of α-pinene that reacted with OH under high $NO_x$ ($0.75 \pm 0.06$) is lower than that under low $NO_x$ ($0.83 \pm 0.04$). These small, but statistically significant differences ($0.08$; t-test, $p=0.004$) in the contribution of ozone/OH to α-pinene oxidation are expected to explain a small part of the differences in SOA yields and chemical composition observed at low and high $NO_x$, with higher fraction of ozonolysis products under high $NO_x$ conditions. Despite this, we expect the influence of $NO_x$ on the fate of $RO_2$ to be the main driver behind the observed differences in SOA yields and chemical composition between low and high $NO_x$ conditions (because

ozonolysis products are only a minor fraction and differences between the two conditions are rather small).

*P12, 9: Please use another word than "corresponding" for these comparisons. It is misleading.*

**Response:** The text was modified as follows:

RH was then changed in the model and the effects of RH on SOA yields and degree of oxygenation were then evaluated.

*P13, 1-10: This is also consistent with more a-pinene producing more SOA and thereby condensation sink (CS), which in turn can more efficiently compete with the walls as a sink of low-volatile vapors. See e.g. the papers cited above. The authors can easily do a sensitivity check for this effect. If there is a large change in particle vs wall loss rates, point 1 should be reconsidered. On the other hand, if the initial seed provides a relatively constant CS (compared to wall losses) for every experiment, then the claim in point 2 that seed concentrations in general are not important seems unjustified, since this was not probed at all in these experiments.*

**Response:** The conclusions in point 2 are based on the fact that we did not observe a correlation between the yields and the initial seed mass or surface concentration, which span a factor of 2.5 (e.g. higher than the span of α-pinene concentration, see point 1). Therefore, would there be an influence of the seed surface we would have observed it, but this was not the case. As mentioned in the manuscript, this is consistent with new results suggesting that for the α-pinene system SOA formation is dominated by quasi-equilibrium growth and vapour losses to the walls do not depend on the seed concentrations, but rather on SOA formation rates (Nah et al., 2016).

In Point 1, we simply describe first the observation that SOA yields increase with α-pinene concentrations. This is expected from the equilibrium growth of α-pinene SOA from condensable semi-volatile vapours, and is consistent with point 2 (this is the basis of the partitioning theory). We have now assessed the change in the particle condensation sink between experiments with α-pinene concentration equal 20 and 30ppb. The change is too negligible, ~10%, to have an impact on the ratio between the particle and wall surfaces and on the vapour wall losses. Therefore, we think that point 1 is still valid.

*P13, 2: Please clearly distinguish between percentages and percentage points. I expect this should be the latter.*

**Response:** It is percentage point, as stated by the reviewer. In the corrected version of the manuscript we add "(percentage point)" after the 2%.

*Fig. 4: There is a clear bimodal distribution for most cases, which is also noted in the manuscript. However, I find the explanations and discussion about it lacking. The authors state that particle number increased, but no new particle formation was observed. This needs some further discussion. Where do the particles come from then? Additionally, the bimodality is used as proof for LLPS, which the model also predicts, but I do not find a clear description of why the organics form this bimodal distribution. Fig. 7&8: Make the contrast between the currently light and full colors more visible. At least on my screen some pairs were hard to distinguish.*

**Response:** The development of a bimodal distribution is described in section 3.3, pages 14 – 15 and then discussed on page 20, lines 14 – 26. The aerosol size-resolved chemical composition is used only to infer the behaviour of the absorptive organic phase and its mixing with the inorganic seed, rather than to model the particle dynamics in the chamber and the processes involved (in our opinion the latter needs much more constraints, which we do not have). As we have mentioned (on page 15, line 2), we have observed an increase in the total particle number, but based on Fig. S14, we did not observe clear and intense nucleation events (substantial increase in the number of small particles). Therefore, while we cannot exclude the occurrence of moderate new particle formation events, we believe that we do not have the adequate tools (detection of new particles below 10 nm) to confirm it. We also cannot exclude that the transmission efficiency of smaller particles in the lines increases upon their growth, which would also explain the increase in the particle number. Accordingly, we prefer keeping the same statements in the text, without explicitly affirming that new particle events occurred, especially that whether nucleation occurred or not does not change in our opinion the conclusions of the paper.

For all experiments, the aerosol size distributions show two externally mixed aerosol populations, with a mode at lower diameters (~200 nm, mode 1) mostly containing SOA and another mode at higher diameters (~400 nm, mode 2) mostly consisting of the seed. As mentioned on page 20, the formation of these two populations may occur by the homogeneous

or heterogeneous nucleation of highly oxidized non-volatile products. Homogeneous nucleation implies new particle formation (which we did not exclude as mentioned above), while heterogeneous nucleation proceeds via condensational growth of non-volatile vapours onto smaller particles (the particle surface mode). Both processes are expected to create small organic rich particles, providing an organic absorptive phase into which additional semi-volatile compounds may preferentially partition. While we cannot exclude any of the two processes (homogeneous or heterogeneous nucleation), the size-resolved chemical composition information is used here to provide compelling evidence that the organic and the electrolyte phases are not mixed, confirming the modelling results.

*P15, 36-37 and related O:C discussion: A variation of 0.03 from 0.45 to 0.48 is considered "almost constant" while an increase of 0.08 is considered significant? And only several pages later in section 4.6 is it noted that the uncertainty in O:C is 20-30%. It is also noted that the O:C values are likely biased low since the latest parametrization for O:C calculations are not used. These things should be mentioned earlier, so a reader can properly assess the meaningfulness of the comparisons done in sections 4.2-4.3. Considering all the above, the tuning of the model to match these values does not in my mind give much more insight into the formation mechanisms of SOA in this system.*

**Response:** In general, while it is hard to assess the uncertainties related to the AMS O:C measurements, a distinction should be made between measurement precision and accuracy. We do not expect the accuracy of the O:C ratios determined by the HR-ToF-AMS to be less than 20%, while changes in the O:C ratios can be detected more precisely by the instrument (~1-2%). As noted by the reviewer, we have used an earlier parameterization to determine the O:C ratio (Aiken et al., 2008), and mentioned that the later parameterization would yield higher O:C values (Canagaratna et al., 2015). While it is not straightforward (impossible) to judge which parameterization is better for our system, based on our recent measurements of ambient samples (see Bozzetti et al. (2017)), the later parameterization does not seem to provide more accurate results. Therefore, we decided to maintain the earlier parameterization.

As suggested by the reviewer, in the corrected version of the manuscript we will discuss earlier in the text the uncertainties related to the determination of the O:C ratios. In the method section (2.1), the following text is added:

The HR-ToF-AMS data were processed and analysed using the analysis software SQUIRREL (SeQUential Igor data RetRiEvaL) v.1.52L and PIKA (Peak Integration by Key Analysis)

v.1.11L for IGOR Pro software package (Wavemetrics, Inc., Portland, OR, USA). From the HR analysis of the mass spectra, the O:C ratios of the bulk OA were determined based on the parameterization proposed by Aiken et al. (2008). We note that while the assessment of the uncertainties related to the O:C measurements by the HR-ToF-AMS is not straightforward, a distinction should be made between measurement precision and accuracy. We do not expect the accuracy of the O:C ratios determined by the HR-ToF-AMS to be less than ~20% (Aiken et al., 2008; Pieber et al., 2016; Canagaratna et al., 2015; Bozzetti et al., 2017). For example, the use of a more recent parameterization (Canagaratna et al., 2015) would yield higher O:C values (by 18%) and the O:C ratios reported here may be regarded as lowest estimates. By contrast, relative changes in the O:C ratios are expected to be detected more precisely by the instrument (~1-2%). The influence of potential biases and uncertainties in the determination of the O:C ratios on our results will be discussed in the text.

The first statement mentioned by the reviewer reads as follows: "This agreement was achieved although the model could not reproduce the high values of nor the change in the O:C ratios with RH (the modelled O:C remained almost constant at 0.45-0.48 at low and high RH, while the measured O:C increased from 0.56 at low RH to 0.64 at high RH)." The point here is that the model underestimated the O:C values by 24%, and predicted a smaller increase in their values with RH (6% compared to 14%). This would be even worse if the more recent parameterization of Canagaratna et al. (2015) was used. The above sentence is modified as follows:

This agreement was achieved although the model underestimated the O:C ratios by 24% and predicted a smaller increase in the O:C values with RH than observed. The modelled O:C remained almost constant at 0.45-0.48 at low and high RH (difference of 6%), while the measured O:C increased from 0.56 at low RH to 0.64 at high RH (change of 14%). We also note that the use of the more recent parameterization by Canagaratna et al. (2015) would yield even higher O:C values, widening the gap between measured and modelled O:C ratios.

The other point raised by the reviewer pertains to the discussion of the differences between the two parameterizations on page 19. The point here is that the use of the most recent parameterization would yield higher O:C ratios, which would require including more fragmented products in the model than already done to reproduce the measured O:C ratios. Therefore, using either parameterization will not change the conclusions of the paper in a qualitative term, suggesting that fragmentation products should be included in the model to reproduce the observations. In the discussion section, we added the following clarifications:

The measured O:C ratio is a key parameter for constraining the model. Here, we have used the high resolution parameterization proposed by Aiken et al. (2007), while the use of the more recent parameterization by Canagaratna et al. (2015) would result in even higher O:C ratios (by 18%). Higher O:C ratios would require increasing even further the contribution or the degree of oxidation of the fragmented compounds and would imply that the model predicts an even higher sensitivity of the yields to RH. Therefore, the O:C values used here yield more conservative estimates of the contributions (or the degree of oxidation) of fragmented products in the model and of the sensitivity of the yields and O:C ratios to the RH.

*P18, 3-5: Is this shown somewhere, or just stated? There were also other places where the formulations are such that I expect there to be a figure showing the result. The authors should consider adding "not shown" in places where the information is not visible in any plot or table.*

**Response:** This is indeed stated and not shown. In the revised version of the manuscript we have added "(not shown)" here and in places where the information is not visible in any plot or table.

*P18, 22-26. There are too many numbers listed in the text, and this is especially true here. Please consider rewriting this.*

**Response:** The section reads as follows: "This increased partitioning is limited by an increase in the activity coefficients of these compounds (for experiment 8 from 1.69 and 1.63 at low RH to 2.70 and 2.93 at high RH and for experiment 13 from 1.49 and 1.54 at low RH to 2.88 and 3.24 at high RH). Conversely, the 5-fold enhanced partitioning of the fragmented and more functionalized compounds (5-COOH-3-OH-pentanal and succinic acid) into the condensed phase at high RH is driven by the increase of the absorptive mass and the slight decrease of the compounds' activity coefficients (for experiment 8 from 0.84 and 0.51 at low RH to 0.68 and 0.43 at high RH and for experiment 13 from 0.90 and 0.56 at low RH to 0.69 and 0.44 at high RH)." We agree that the text contains a lot of numbers, but all of them are between brackets, so that the reader is not obliged to read them. We think these numbers are necessary to provide an idea about the range of the modelled activity coefficients, under different RH and for different types of compounds.

*P18, 19-22 and Fig. 9: It is hard to follow discussion about increase of factors 2-5 from a plot ranging 14 orders of magnitude. Could Fig. 9 be moved to the SI, and some more specific plot included in the main text?*

**Response:** Figure 9 shows the partitioning of the individual model compounds to the different phases, which provides an idea which compounds are expected to contribute to the two particle phases under different conditions. We think this figure is required in the main text.

*P18, 39-40: I do not understand at all what this sentence is supposed to say.*

**Response:** This sentence has been removed from the manuscript.

*P19, 7: "sufficient" for what?*

**Response:** We specify "sufficient to represent the species within one volatility bin".

*P20, 22: Expected based on what?*

**Response:** This section has been reformulated as follows:

Considering the size-resolved particle chemical composition discussed in Sect. 3.3 (Figure 4), LLPS is likely not realized within single particles but the aerosol population splits up into a predominantly organic mode at ~200 nm and a predominantly inorganic mode at ~400 nm. The formation of these two populations may occur by the homogeneous or heterogeneous nucleation of highly oxidized non-volatile products. Homogeneous nucleation implies new particle formation (which would only be moderate– see Figures S13 and S14 – due to the high condensation sink in the chamber), while heterogeneous nucleation proceeds via condensational growth (which would occur on smaller particles with higher surface). Both processes are expected to create small organic rich particles, providing an organic absorptive phase into which additional semi-volatile compounds may preferentially partition. When the organic and electrolyte phases are present in different particles the two phases communicate via gas phase diffusion, and equilibration time-scales depend on the components' volatility. For compounds with $C^* = 0.1\text{-}100$ µg m$^{-3}$ equilibration occurs within time-scales of minutes to tens of minutes, assuming no bulk phase diffusion limitations (Marcolli et al., 2004). In the larger particle electrolyte-rich mode, the inorganic ions would exert a salting-out effect driving the organic compounds to partition to the gas-phase or into the smaller organic-rich

particles. This would prevent the organic compounds from partitioning in significant amounts into the seed aerosol from the beginning and deplete even further these larger particles from the organic material. Under such a scenario an externally-mixed phase-separated aerosol may evolve in the smog chamber. Overall, the size-resolved chemical composition information confirms the modelling results providing compelling evidence for organic-electrolyte LLPS.

*P21, 28-29: This is too strong a statement in my opinion. Rather say that only with inclusion of the fragments could your model describe both SOA mass and O:C.*

**Response:** This has been modified in the revised version of the manuscript as suggested by the reviewer:

Our results show that only with the inclusion of fragmentation products could the model simultaneously explain SOA concentrations and O:C ratios.

*P21, 40-P22, 2: Such a statement should be included much earlier in the discussions on bimodality, and not saved to the last lines of the manuscript.*

**Response:** This statement has been already discussed earlier in the manuscript on page 20 and modified in the new version of the text based on an earlier comment of the reviewer. The section on page 20 now reads as follows:

Considering the size-resolved particle chemical composition discussed in Sect. 3.3 (Figure 4), LLPS is likely not realized within single particles but the aerosol population splits up into a predominantly organic mode at ~200 nm and a predominantly inorganic mode at ~400 nm. The formation of these two populations may occur by the homogeneous or heterogeneous nucleation of highly oxidized non-volatile products. Homogeneous nucleation implies new particle formation (which would only be moderate– see Figures S13 and S14 – due to the high condensation sink in the chamber), while heterogeneous nucleation proceeds via condensational growth (which would occur on smaller particles with higher surface). Both processes are expected to create small organic rich particles, providing an organic absorptive phase into which additional semi-volatile compounds may preferentially partition. When the organic and electrolyte phases are present in different particles the two phases communicate via gas phase diffusion, and equilibration time-scales depend on the components' volatility. For compounds with $C^* = 0.1\text{-}100 \ \mu g \ m^{-3}$ equilibration occurs within time-scales of minutes to tens of minutes, assuming no bulk phase diffusion limitations (Marcolli et al., 2004). In the

larger particle electrolyte-rich mode, the inorganic ions would exert a salting-out effect driving the organic compounds to partition to the gas-phase or into the smaller organic-rich particles. This would prevent the organic compounds from partitioning in significant amounts into the seed aerosol from the beginning and deplete even further these larger particles from the organic material. Under such scenario an externally-mixed phase-separated aerosol may evolve in the smog chamber. Overall, the size-resolved chemical composition information confirms the modelling results providing compelling evidence for organic-electrolyte LLPS.

*P22, 2: Again, what is this expectation based on? Work in this paper or other work?*

**Response:** This is based on our work and has been clarified in the new version of the manuscript.

**References**

[revised manuscript text omitted]